# OFFLINE RL IN REGULAR DECISION PROCESSES: SAMPLE EFFICIENCY VIA LANGUAGE METRICS

**Ahana Deb**[1*]  **Roberto Cipollone**[2†]  **Anders Jonsson**[1]  **Alessandro Ronca**[3]
**Mohammad Sadegh Talebi**[4]
[1]Universitat Pompeu Fabra  [2]Leonardo S.p.A.  [3]University of Oxford
[4]University of Copenhagen

## ABSTRACT

This work studies offline Reinforcement Learning (RL) in a class of non-Markovian environments called Regular Decision Processes (RDPs). In RDPs, the unknown dependency of future observations and rewards from the past interactions can be captured by some hidden finite-state automaton. For this reason, many RDP algorithms first reconstruct this unknown dependency using automata learning techniques. In this paper, we consider episodic RDPs and show that it is possible to overcome the limitations of existing offline RL algorithms for RDPs via the introduction of two original techniques: a novel metric grounded in formal language theory and an approach based on Count-Min-Sketch (CMS). Owing to the novel language metric, our algorithm is proven to be more sample efficient than existing results, and in some problem instances admitting low complexity languages, the gain is showcased to be exponential in the episode length. The CMS-based approach removes the need for naïve counting and alleviates the memory requirements for long planning horizons. We derive Probably Approximately Correct (PAC) sample complexity bounds associated to each of these techniques, and validate the approach experimentally.

## 1 INTRODUCTION

The Markov assumption is fundamental for most Reinforcement Learning (RL) algorithms, requiring that the immediate reward and transition only depend on the last observation and action. Thanks to this property, computing (near-)optimal policies involves only functions over observations and actions. However, in complex environments, observations may not be complete representations of the internal environment state. In this work, we consider RL in Non-Markov Decision Processes (NMDPs) (Whitehead & Lin, 1995; Bacchus et al., 1997), expressive models where the probability of future observations and rewards may depend on the entire history, i.e. the past interaction sequence composed of observations and actions. However, the unrestricted dynamics of NMDPs make them intractable from both statistical and computational standpoints. This has steered much research effort towards tractable subclasses of NMDPs. In this work, we focus on Regular Decision Processes (RDPs) (Brafman & De Giacomo, 2019; 2024). In RDPs, the distribution of the next observation and reward is determined by conditions over the history that, in terms of formal language theory, fall within the class of the regular languages. This is a rich class containing many fundamental temporal patterns, including all patterns that can be specified in Linear Temporal Logic (Manna & Pnueli, 1989; De Giacomo & Vardi, 2013) and those captured by Reward Machines (Toro Icarte et al., 2018; 2019; Toro Icarte et al., 2022). Thus, RDPs can model complex temporal dependencies that may be based on events that occur arbitrarily far in the past, e.g. that an agent may only enter a restricted area if it has previously asked for permission and the access was granted. At the same time, RDPs enjoy many favourable properties, which can be leveraged to develop effective RL algorithms. Prominently, the fact that the dynamics of an RDP can be represented by a probabilistic-deterministic finite automaton. This key fact is the basis of all existing RL algorithms for RDPs: learning the automaton underlying an RDP amounts to *learning a representation of the histories* that is instrumental to tackle non-Markovianity, as it allows one to work with an associated Markov Decision Process (MDP).

---

*Correspondence to: `ahana.deb@upf.edu`.
†The work was conducted while the author was at Sapienza University of Rome.

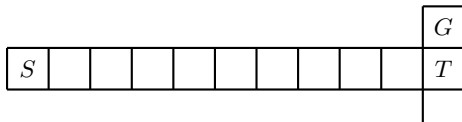

Figure 1: T-maze (Bakker, 2001) with corridor length $N = 10$. The observation produced at the initial position $S$ indicates the position of the goal $G$ at the end of the corridor for the current episode.

It is worth noting that RDPs are related to decision processes operating under partial observability. More precisely, the automaton state of an RDP can be seen as a hidden information state (Subramanian et al., 2022), and as shown by Brafman & De Giacomo (2019), any RDP is also a POMDP (Kaelbling et al., 1998), whose hidden dynamics evolve according to its finite-state automaton. Thus RDPs constitute the class of POMDPs where hidden states are determined by the history of observations, namely, those in which the agent can have sharp beliefs. They also fall into the framework of Predictive State Representations (PSRs) (Bowling et al., 2006). Therefore, in the hierarchy of decision processes, RDPs greatly generalise MDPs, without introducing strong forms of partial observability present in more general processes, such as POMDPs, which quickly lead to intractability (Papadimitriou & Tsitsiklis, 1987). Despite such applicability, provably correct and sample efficient algorithms for RDPs are still missing. On one hand, local optimisation approaches are computationally efficient, but lack correctness guarantees, e.g. Abadi & Brafman (2020), Toro Icarte et al. (2019) and all RL algorithms with policy networks that can operate on sequences. On the other hand, algorithms with formal guarantees do not provide a practical implementation (Cipollone et al., 2023; Ronca & De Giacomo, 2021) or can only be applied effectively in small environments (Ronca et al., 2022).

We study offline RL in episodic RDPs, where the goal is to find a near-optimal policy from a dataset pre-collected using a behavior policy. A recent algorithm called RegORL (Cipollone et al., 2023) achieves a sample complexity with polynomial dependency on the episode length and the number of actions, observations, and automaton states. Besides, the sample complexity inversely depends on a complexity notion called $L_\infty^p$-*distinguishability*, which is grounded in automata theory and captures the ability to distinguish between relevant histories generated by the behavior policy. This sample complexity bound exhibits two major drawbacks. First, the $L_\infty^p$-distinguishability could be very small relative to problem parameters, and there exist RDPs in which $L_\infty^p$-distinguishability $\simeq 2^{-H}$, for episode length $H$, as formalized in Theorem 1, which asserts that the sample complexity of RegORL can be exponential in $H$. Indeed, such an exponential blow-up is observed in many domains that seem simple from an intuitive point of view, such as T-maze by Bakker (2001), introduced below and discussed throughout the paper. Second, the only existing sample complexity lower bound presented in Cipollone et al. (2023) depends on $L_1^p$-distinguishability, another complexity notion that is larger than $L_\infty^p$-distinguishability and does not suffer from the exponential decay above. Thus, the following question naturally arises: *Is it possible to provably learn a near-optimal policy in RDPs offline without requiring an exponential (in $H$) sample complexity, while admitting a computationally tractable implementation?* We answer this question in the affirmative by developing novel algorithmic ideas that build on the theory of formal languages as well as using a suitable data structure.

**Example 1** (T-maze). As a running example, we consider a variant of T-Maze(c) (Bakker, 2001) that we call T-maze. An agent has to reach a goal $G$ from an initial position $S$ in a corridor of length $N$ that terminates with a T-junction as shown in Figure 1, using actions $North$, $South$, $East$, and $West$. In each episode, the rewarding goal $G$ is placed above or below the T-junction uniformly at random. The initial observation of the agent is $011$ if the goal is above the T-junction, and $110$ otherwise. At every step in the corridor the observation is $101$ or $111$ with equal probability, and at the T-junction and beyond the observation is $010$. In particular, when crossing the corridor, the agent cannot observe its precise location or the goal position. This yields history-dependent dynamics that cannot be modeled by an MDP or any k-order MDP. As we show later, this domain can be expressed as an RDP.

**Contributions** Our main contribution is a provably sample efficient algorithm for offline RL in episodic RDPs. At the core is a statistical test defined via a novel language metric $L_{\mathcal{X}}$, whose design borrows ideas from the theory of formal languages. Its construction involves a sophisticated way of expressively capturing relevant patterns on episode traces in the collected data, which allows for defining a suitable hierarchy of language families that proves effective for learning in RDPs. Specifically, as formalized in Theorem 3, this yields a PAC sample complexity bound that depends on a new notion of complexity called $L_{\mathcal{X}}$-*distinguishability*, which offers a more refined notion than $L_\infty^p$-distinguishability and, importantly, does not suffer from an exponential blow-up in domains with a low language-theoretic complexity. Second, we develop a way to compactly represent probability

distributions on large sets in order to mitigate the memory requirements of RL algorithms that operate on large sets of histories. We develop a solution that uses the data structure Count-Min-Sketch (CMS) (Cormode & Muthukrishnan, 2005), taking inspiration from the automaton-learning algorithm FlexFringe (Baumgartner & Verwer, 2023). To validate our claims, we provide a theoretical analysis for both variants, when the $L_{\mathcal{X}}$-distinguishability or CMS is used. Finally, we report numerical experiments to demonstrate the empirical performance of the proposed approaches.

## 1.1 RELATED WORK

There is a rich and growing literature on offline (Markovian) RL that covers a broad range of MDP settings; e.g., (Chen & Jiang, 2019; Jin et al., 2021; Li et al., 2024; Rashidinejad et al., 2021; Ren et al., 2021; Uehara & Sun, 2022; Uehara et al., 2022; Xie et al., 2021; Yin & Wang, 2021; Zhan et al., 2022; Sun et al., 2023). In particular, it is by now well established that the optimal sample complexity in the episodic setting depends on the size of state-space, episode length, as well as some notion of concentrability, reflecting the distribution mismatch between the behavior and optimal policies, and optimal algorithms are reported (Li et al., 2024).

**Non-Markovian RL**  The first online RL algorithm for RDPs is provided in Abadi & Brafman (2020). Ronca & De Giacomo (2021) and Ronca et al. (2022) developed the first online RL algorithms with sample complexity guarantees, adapting analogous results from automata learning literature (Balle, 2013; Balle et al., 2013; 2014; Clark & Thollard, 2004; Palmer & Goldberg, 2007; Ron et al., 1998). Cipollone et al. (2023) introduced RegORL, a provably efficient algorithm for offline RL in RDPs. We study the same setting and improve on two weaknesses of RegORL regarding the sample complexity and the space requirements. RL with non-Markovian rewards have been studied via Reward Machines (Bourel et al., 2023; 2024; Giacomo et al., 2020; Hasanbeig et al., 2021; Toro Icarte et al., 2018; Xu et al., 2020). Existing literature, however, provide theoretical sample efficiency when the automaton of such processes is known, which makes the Markovian state computable; e.g., the online RL algorithm in Toro Icarte et al. (2019) could be applied to RDPs, but it is not proven to be sample efficient. Lastly, non-Markovianity is also introduced by the logical specifications that the agent is required to satisfy (Bozkurt et al., 2020; Fu & Topcu, 2014; Hahn et al., 2019; Hammond et al., 2021; Hasanbeig et al., 2020); however, it is resolved a priori from the known specifications.

**POMDPs, PSRs, and State Representation**  As discussed, RDPs are also POMDPs, and any RL algorithm for POMDPs can be applied to RDPs. Tractable learning in POMDPs, to our best knowledge, has been studied for special cases, such as undercomplete POMDPs (Guo et al., 2022; Jin et al., 2020), few-step reachability (Guo et al., 2016), ergodicity (Azizzadenesheli et al., 2016), few-step decodability (Efroni et al., 2022; Krishnamurthy et al., 2016), or weakly-revealing (Liu et al., 2022). Hahn et al. (2024) further explore Omega-Regular Decision Processes (ODPs) which generalizes RDPs with an $\omega$-regular lookahead, and present classical complexity results. However, none of these fully capture the entire RDP class. Non-Markovian dynamics, and thus RDPs, fall into the framework of PSRs (Bowling et al., 2006; James & Singh, 2004; Kulesza et al., 2015; Singh et al., 2003). However, existing sample complexity bounds for online RL in PSRs (Zhan et al., 2023) involve PSR-specific parameters that do not immediately apply to RDPs. RDPs bear resemblance to state representation methods (Mahmud, 2010; Maillard et al., 2013; Ortner et al., 2019; Lattimore et al., 2013), since they provide a compact way of mapping from histories to a state space via their underlying automata. However, existing bounds for state representations grow linearly in the number of representations, which is exponential in the number of states in our case.

## 2 PRELIMINARIES

**Notation**  Given a set $\mathcal{Y}$, $\Delta(\mathcal{Y})$ denotes the set of probability distributions over $\mathcal{Y}$. For a function $f : \mathcal{X} \to \Delta(\mathcal{Y})$, $f(y \mid x)$ is the probability of $y \in \mathcal{Y}$ given $x \in \mathcal{X}$. Further, we write $y \sim f(x)$ to abbreviate $y \sim f(\cdot \mid x)$. Given an event $E$, $\mathbb{I}(E)$ denotes the indicator function of $E$, which equals 1 if $E$ is true, and 0 otherwise. For any pair of integers $m$ and $n$ such that $0 \le m \le n$, we let $[\![m, n]\!] := \{m, \dots, n\}$ and $[\![n]\!] := [\![1, n]\!]$. The notation $\widetilde{\mathcal{O}}(\cdot)$ hides poly-logarithmic terms.

**Count-Min-Sketch**  Count-Min-Sketch, or CMS (Cormode & Muthukrishnan, 2005), is a data structure that compactly represents a large non-negative vector $v = (v_1, \dots, v_m)$. CMS takes two

parameters $\delta_c$ and $\varepsilon$ as input, and constructs a matrix $C$ with $d = \lceil \log \frac{1}{\delta_c} \rceil$ rows and $w = \lceil \frac{e}{\varepsilon} \rceil$ columns. For each row $j \in [\![d]\!]$, CMS picks a hash function $h_j : [\![m]\!] \to [\![w]\!]$ uniformly at random from a pairwise independent family (Motwani & Raghavan, 1995). Initially, all elements of $v$ and $C$ equal 0. An update $(i, c)$ consists in incrementing the element $v_i$ by $c > 0$. CMS approximates an update $(i, c)$ by incrementing $C(j, h_j(i))$ by $c$ for each $j \in [\![d]\!]$. At any moment, a point query $\widetilde{v}_i$ returns an estimate of $v_i$ by taking the minimum of the row estimates, i.e. $\widetilde{v}_i = \min_j C(j, h_j(i))$. For each row $j \in [\![d]\!]$, since the updates for $v_i$ are always stored in the element $C(j, h_j(i))$, it follows that $C(j, h_j(i)) \geq v_i$. Hence each point query satisfies $\widetilde{v}_i \geq v_i$, i.e. CMS never underestimates $v_i$.

## 2.1 Languages and Operators

An *alphabet* $\Gamma$ is a finite non-empty set of elements called *letters*. A *string* $x$ over $\Gamma$ is a concatenation $a_1 \cdots a_\ell$ of letters from $\Gamma$; we call $\ell$ its length, and we write $|x| = \ell$. In particular, the string containing no letters, having length zero, is a valid string called the *empty string*, and it is denoted by $\lambda$. Given two strings $x = a_1 \cdots a_\ell$ and $y = b_1 \cdots b_m$, their concatenation $xy$ is the string $a_1 \cdots a_\ell b_1 \cdots b_m$. In particular, $x\lambda = \lambda x = x$. Note that $|xy| = |x| + |y|$. Concatenation is associative and hence we can write the concatenation $x_1 x_2 \cdots x_k$ of an arbitrary number of strings. The set of all strings over alphabet $\Gamma$ is written as $\Gamma^*$, and the set of all strings of length $\ell$ is written as $\Gamma^\ell$. Thus, $\Gamma^* = \cup_{\ell \in \mathbb{N}} \Gamma^\ell$. A *language* is a subset of $\Gamma^*$. Given two languages $X_1$ and $X_2$, their concatenation is the language defined by $X_1 X_2 = \{x_1 x_2 \mid x_1 \in X_1, x_2 \in X_2\}$. When concatenating with a singleton language $\{x\}$, we often write $Xx$ instead of $X\{x\}$, and $xX$ instead of $\{x\}X$. Concatenation of languages is associative, allowing us to write the concatenation $X_1 X_2 \cdots X_k$ of an arbitrary number of languages.

Given the fundamental definitions above, we introduce a novel operator to construct sets of languages, which will play a key role in the algorithm design through defining relevant patterns on episode traces. It is inspired by classes of languages in the first level of the *dot-depth hierarchy*, a well-known hierarchy of star-free regular languages (Simon, 1972; Thérien, 2005; Pin, 2017).

**Definition 1.** For $\ell \in \mathbb{N}$ and $k \in [\![\ell]\!]$, the operator $\mathrm{C}_k^\ell$ maps any set of languages $\mathcal{G}$ to a new set of languages as follows:

$$\mathrm{C}_k^\ell(\mathcal{G}) = \left\{ \{x_0 G_1 \cdots x_{k-1} G_k x_k \mid x_0, \ldots, x_k \in \Gamma^*, |x_0 \cdots x_k| = (\ell - k)\} \mid G_1, \ldots, G_k \in \mathcal{G} \right\}.$$

In the definition, each $x_i$ can be any string over the chosen alphabet $\Gamma$, including the empty string $\lambda$. Thus, $x_i$ are arbitrary strings between a string from $G_i$ and the next string from $G_{i+1}$. Each resulting string consists of $\ell$ occurrences of a letter from $\Gamma$ or a string from $G_1, \ldots, G_k \in \mathcal{G}$. Different choices of $G_1, \ldots, G_k \in \mathcal{G}$ yield different languages in the resulting set.

**Example 2.** Let $\Gamma = \{a, b, c\}$ and let $\mathcal{G} = \{\{a\}, \{b\}\}$. Then

$$\mathrm{C}_1^{10}(\mathcal{G}) = \left\{ \{xay \mid x, y \in \Gamma^*, |xy| = 9\}, \{xby \mid x, y \in \Gamma^*, |xy| = 9\} \right\}$$

is the set consisting of (i) the language of all 10-letter strings that include an occurrence of $a$, and (ii) the language of all 10-letter strings that include an occurrence of $b$. Similarly,

$$\mathrm{C}_2^{10}(\mathcal{G}) = \left\{ \{xayaz \mid x, y, z \in \Gamma^*, |xyz| = 8\}, \{xbybz \mid x, y, z \in \Gamma^*, |xyz| = 8\}, \right.$$
$$\left. \{xaybz \mid x, y, z \in \Gamma^*, |xyz| = 8\}, \{xbyaz \mid x, y, z \in \Gamma^*, |xyz| = 8\} \right\}$$

is the set consisting of (i) the language of all 10-letter strings that include two occurrences of $a$, (ii) the language of all 10-letter strings that include two occurrences of $b$, (iii) the language of all 10-letter strings that include an occurrence of $a$ followed by an occurrence of $b$, (iv) the language of all 10-letter strings that include an occurrence of $b$ followed by an occurrence of $a$.

## 2.2 Episodic Regular Decision Processes

A generic episodic decision process is a tuple $\mathbf{P} = \langle \mathcal{O}, \mathcal{A}, \mathcal{R}, \bar{T}, \bar{R}, H \rangle$, where $\mathcal{O}$ is a finite set of observations, $\mathcal{A}$ is a finite set of actions, $\mathcal{R} \subset [0, 1]$ is a finite set of rewards, and $H \geq 1$ is a finite horizon. We frequently consider the concatenation $\mathcal{AO}$ of the sets $\mathcal{A}$ and $\mathcal{O}$. Let $\mathcal{H}_t = (\mathcal{AO})^{t+1}$ be the set of histories of length $t + 1$, and let $e_{m:n} \in \mathcal{H}_{n-m}$ denote a history from time $m$ to time $n$, both included. Each action-observation pair $ao \in \mathcal{AO}$ in a history has an associated reward label $r \in \mathcal{R}$, which we write $ao/r \in \mathcal{AO}/\mathcal{R}$ with the understanding that the slash corresponds to string concatenation. A *trajectory* $e_{0:T}$ is the full history generated until (and including) time $T$.

We assume that a trajectory $e_{0:T}$ can be partitioned into *episodes* $e_{\ell:\ell+H} \in \mathcal{H}_H$ of length $H + 1$. In each episode $e_{0:H}$, $a_0 = a_\perp$ is a dummy action used to initialize the distribution on $\mathcal{H}_0$. The transition function $\bar{T} : \mathcal{H} \times \mathcal{A} \to \Delta(\mathcal{O})$ and the reward function $\bar{R} : \mathcal{H} \times \mathcal{A} \to \Delta(\mathcal{R})$ depend on the current history in $\mathcal{H} = \cup_{t=0}^{H} \mathcal{H}_t$. Given $\mathbf{P}$, a generic policy is a function $\pi : (\mathcal{AO})^* \to \Delta(\mathcal{A})$ that maps trajectories to distributions over actions. The value function $V^\pi : [\![0, H]\!] \times \mathcal{H} \to \mathbb{R}$ of a policy $\pi$ is a mapping that assigns real values to histories. For $h \in \mathcal{H}$, it is defined as $V^\pi(H, h) := 0$ and

$$V^\pi(t, h) := \mathbb{E}\left[\sum_{i=t+1}^{H} r_i \,\middle|\, h, \pi\right], \quad \forall t < H, \forall h \in \mathcal{H}_t. \tag{1}$$

For brevity, we write $V_t^\pi(h) := V^\pi(t, h)$. The optimal value function $V^*$ is defined as $V_t^*(h) := \sup_\pi V_t^\pi(h), \forall t \in [\![0, H]\!], \forall h \in \mathcal{H}_t$, where sup is taken over all policies $\pi : (\mathcal{AO})^* \to \Delta(\mathcal{A})$. Any policy achieving $V^*$ is called optimal, which we denote by $\pi^*$; namely $V^{\pi^*} = V^*$. In what follows, we consider simpler policies of the form $\pi : \mathcal{H} \to \Delta(\mathcal{A})$ mapping finite histories to distributions over actions. Let $\Pi_\mathcal{H}$ denote the set of such policies. It can be shown that $\Pi_\mathcal{H}$ always contains an optimal policy, i.e. $V_t^*(h) := \max_{\pi \in \Pi_\mathcal{H}} V_t^\pi(h), \forall t \in [H], \forall h \in \mathcal{H}_t$. A policy $\hat{\pi}$ is $\varepsilon$-optimal iff $\mathbb{E}_{h_0}[V_0^*(h_0) - V_0^{\hat{\pi}}(h_0)] \leq \varepsilon$, where $h_0 = a_\perp o_0$, for some random $o_0 \in \mathcal{O}$.

**Episodic RDPs** An episodic Regular Decision Process (RDP) (Abadi & Brafman, 2020; Brafman & De Giacomo, 2019; 2024) is an episodic decision process $\mathbf{R} = \langle \mathcal{O}, \mathcal{A}, \mathcal{R}, \bar{T}, \bar{R}, H \rangle$ described by a *finite transducer* (Moore machine) $\langle \mathcal{U}, \Sigma, \Omega, \tau, \theta, u_0 \rangle$, where $\mathcal{U}$ is a finite set of states, $\Sigma = \mathcal{AO}$ is a finite input alphabet composed of actions and observations, $\Omega$ is a finite output alphabet, $\tau : \mathcal{U} \times \Sigma \to \mathcal{U}$ is a transition function, $\theta : \mathcal{U} \to \Omega$ is an output function, and $u_0 \in \mathcal{U}$ is a fixed initial state (Moore, 1956; Shallit, 2008). The output space $\Omega = \Omega_\mathsf{o} \times \Omega_\mathsf{r}$ consists of a finite set of functions that compute the conditional probabilities of observations and rewards, on the form $\Omega_\mathsf{o} \subset \mathcal{A} \to \Delta(\mathcal{O})$ and $\Omega_\mathsf{r} \subset \mathcal{A} \to \Delta(\mathcal{R})$. For simplicity, we use two output functions, $\theta_\mathsf{o} : \mathcal{U} \times \mathcal{A} \to \Delta(\mathcal{O})$ and $\theta_\mathsf{r} : \mathcal{U} \times \mathcal{A} \to \Delta(\mathcal{R})$, to denote the individual conditional probabilities. Let $\tau^{-1}$ denote the inverse of $\tau$, i.e. $\tau^{-1}(u) \subseteq \mathcal{U} \times \mathcal{AO}$ is the subset of state-symbol pairs that map to $u \in \mathcal{U}$. An RDP $\mathbf{R}$ implicitly represents a function $\bar{\tau} : \mathcal{H} \to \mathcal{U}$ from histories in $\mathcal{H}$ to states in $\mathcal{U}$, recursively defined as $\bar{\tau}(h_0) := \tau(u_0, a_0 o_0)$, where $a_0$ is some fixed starting action, and $\bar{\tau}(h_t) := \tau(\bar{\tau}(h_{t-1}), a_t o_t)$. The transition function and reward function of $\mathbf{R}$ are defined as $\bar{T}(o \mid h, a) = \theta_\mathsf{o}(o \mid \bar{\tau}(h), a)$ and $\bar{R}(r \mid h, a) = \theta_\mathsf{r}(r \mid \bar{\tau}(h), a), \forall h \in \mathcal{H}, \forall ao/r \in \mathcal{AO}/\mathcal{R}$. As in previous work (Cipollone et al., 2023), we assume that any episodic RDP generates a designated termination observation $o_\perp \in \mathcal{O}$ after exactly $H$ transitions. This ensures that any episodic RDP is acyclic, i.e. the states can be partitioned as $\mathcal{U} = \mathcal{U}_0 \cup \cdots \cup \mathcal{U}_{H+1}$, where each $\mathcal{U}_{t+1}$ is the set of states generated by the histories in $\mathcal{H}_t$ for each $t \in [\![0, H]\!]$. An RDP is minimal if its Moore machine is minimal. We use $A, O, R, U$ to denote the cardinality of $\mathcal{A}, \mathcal{O}, \mathcal{R}, \mathcal{U}$, respectively, and assume $H \geq 2$, $A \geq 2$ and $O \geq 2$.

Since the conditional probabilities of observations and rewards are fully determined by the current state-action pair $(u, a)$, an RDP $\mathbf{R}$ adheres to the Markov property over its states, but *not over the observations*. Given a state $u_t \in \mathcal{U}$ and an action $a_t \in \mathcal{A}$, the probability of the next transition is

$$\mathbb{P}(r_t, o_t, u_{t+1} \mid u_t, a_t, \mathbf{R}) = \theta_\mathsf{r}(r_t \mid u_t, a_t)\, \theta_\mathsf{o}(o_t \mid u_t, a_t)\, \mathbb{I}(u_{t+1} = \tau(u_t, a_t o_t)).$$

Since RDPs are Markovian in the unobservable states $\mathcal{U}$, there is an important class of policies that is called *regular*. Given an RDP $\mathbf{R}$, a policy $\pi : \mathcal{H} \to \Delta(\mathcal{A})$ is called *regular* if $\pi(h_1) = \pi(h_2)$ whenever $\bar{\tau}(h_1) = \bar{\tau}(h_2)$, for all $h_1, h_2 \in \mathcal{H}$. Hence, we can compactly define a regular policy as a function of the RDP state, i.e. $\pi : \mathcal{U} \to \Delta(\mathcal{A})$. Let $\Pi_\mathbf{R}$ denote the set of regular policies for $\mathbf{R}$. Regular policies exhibit powerful properties. First, under a regular policy, suffixes have the same probability of being generated for histories that map to the same RDP state. Second, there exists at least one optimal policy that is regular. Finally, in the special case where an RDP is Markovian in both observations and rewards, it reduces to a nonstationary episodic MDP.

**Example 3** (RDP for T-maze). Consider the T-maze described in Example 1, for a generic corridor length $N$ and horizon $H = N + 2$. This can be modeled as an episodic RDP $\langle \mathcal{U}, \Sigma, \Omega, \tau, \theta, u_0 \rangle$ with states $\mathcal{U} := u_0 \cup (\{u_{1,\top}, \ldots, u_{N+3,\top}\} \cup \{u_{1,\perp}, \ldots, u_{N+3,\perp}\}) \times \{1, \ldots, H\}$, which include the initial state $u_0$ and two parallel components, $\top$ and $\perp$, for the $N + 3$ cells of the grid world. In addition, each state also includes a counter for the time step. Within each component $\{(u_{i,\top}, t)\}_i$ and $\{(u_{i,\perp}, t)\}$, the transition function $\tau$ mimics the grid world dynamics of the maze and increments the counter $t$. From the initial state and the start action $a_0$, $\tau(u_0, a_0 o_0)$ equals $u_{1,\top}$ if $o_0 = 011$, and $u_{1,\perp}$ if $o_0 = 110$. In the initial state, observations are $\theta_\mathsf{o}(u_0, a_0) = \mathrm{unif}\{110, 011\}$; in the

corridor $\theta_o(u_{i,g}, a) = \operatorname{unif}\{101, 111\}$ for every $i \in [\![N]\!]$ and $g \in \{\top, \bot\}$; and in the T-junction the observation is $010$. The rewards are null, except for a $1$ in the top right or bottom right cell, depending on if the current state is in the component $\top$ or $\bot$, respectively.

**Distinguishability** Consider a minimal RDP $\mathbf{R}$ with states $\mathcal{U} = \cup_{t \in [\![0, H+1]\!]} \mathcal{U}_t$. Given a regular policy $\pi \in \Pi_{\mathbf{R}}$ and $t \in [\![0, H]\!]$, each RDP state $u \in \mathcal{U}_{t+1}$ defines a unique probability distribution $\mathbb{P}(\cdot \mid u_{t+1} = u, \pi)$ on episode suffixes in $\Gamma^{H-t}$, where $\Gamma = \mathcal{AO}/\mathcal{R}$ is the alphabet of action-observation-reward triplets. The states in $\mathcal{U}_{t+1}$ can be compared in terms of the probability distributions they induce over $\Gamma^{H-t}$. Consider any $L = \{L_\ell\}_{\ell=0}^H$, where each $L_\ell$ is a metric over $\Delta(\Gamma^\ell)$. We define the $L$-*distinguishability* of $\mathbf{R}$ under $\pi$ as the maximum $\mu_0 \geq 0$ such that, for any $t \in [\![0, H]\!]$ and any two distinct $u, u' \in \mathcal{U}_{t+1}$, the probability distributions over suffix traces $e_{t+1:H} \in \Gamma^{H-t}$ from the two states satisfy

$$L_{H-t}\big(\mathbb{P}(e_{t+1:H} \mid u_{t+1} = u, \pi), \mathbb{P}(e_{t+1:H} \mid u_{t+1} = u', \pi)\big) \geq \mu_0 \,.$$

We will often omit the remaining episode length $\ell = H - t$ from $L_\ell$ and simply write $L$. We consider the $L_\infty^p$-distinguishability, instantiating the definition above with the metric $L_\infty^p(p_1, p_2) = \max_{j \in [\![\ell]\!], e \in \Gamma^j} |p_1(e*) - p_2(e*)|$, where $p_i(e*)$, $i \in \{1, 2\}$, represents the probability of the trace prefix $e \in \Gamma^j$, followed by any trace $e' \in \Gamma^{\ell-j}$. The $L_1^p$-distinguishability is defined analogously using $L_1^p(p_1, p_2) = \max_{j \in [\![\ell]\!]} \sum_{e \in \Gamma^j} |p_1(e*) - p_2(e*)|$.

## 2.3 OFFLINE RL IN EPISODIC RDPS

Consider a batch dataset $\mathcal{D}$ comprising episodes sampled using a regular behavior policy $\pi^b$. Specifically, the $k$-th episode (or episode trace) in $\mathcal{D}$ is of the form $e_{0:H}^k = a_0^k o_0^k / r_0^k \cdots a_H^k o_H^k / r_H^k$ where, for each $t \in [\![0, H]\!]$,

$$u_0^k = u_0, \quad a_t^k \sim \pi^b(u_t^k), \quad o_t^k \sim \theta_o(u_t^k, a_t^k), \quad r_t^k \sim \theta_r(u_t^k, a_t^k), \quad u_{t+1}^k = \tau(u_t^k, a_t^k o_t^k).$$

The learner seeks an $\varepsilon$-optimal policy $\widehat{\pi}$ for a given accuracy $\varepsilon \in (0, H]$, using the smallest dataset $\mathcal{D}$ possible, without further exploration. More precisely, we aim at finding $\widehat{\pi}$ satisfying $V_0^*(h) - V_0^{\widehat{\pi}}(h) \leq \varepsilon$ for each $h \in \mathcal{H}_0$ with probability at least $1 - \delta$, using the smallest dataset $\mathcal{D}$ possible. We stress that in so doing $\pi^b$ and underlying RDP states $u_t^k$ are unknown to the learner. It suffices to restrict attention to regular $\varepsilon$-optimal policies (cf. Proposition 5 in Appendix B). However, some assumptions must be imposed on $\pi^b$ to provably guarantee that an $\varepsilon$-optimal regular policy can be learnt from $\mathcal{D}$. We first recall the notion of occupancy. Given a regular policy $\pi : \mathcal{U} \to \Delta(\mathcal{A})$ and a time step $t \in [\![0, H]\!]$, let $d_t^\pi \in \Delta(\mathcal{U}_t \times \mathcal{AO})$ be the induced *occupancy*, i.e. a probability distribution over the states in $\mathcal{U}_t$ and the input symbols in $\mathcal{AO}$, recursively defined as $d_0^\pi(u_0, a_0 o_0) = \theta_o(o_0 \mid u_0, a_0)$ and

$$d_t^\pi(u_t, a_t o_t) = \sum_{(u, ao) \in \tau^{-1}(u_t)} d_{t-1}^\pi(u, ao) \cdot \pi(a_t \mid u_t) \cdot \theta_o(o_t \mid u_t, a_t), \quad t > 0.$$

Of particular interest is the occupancy $d_t^* := d_t^{\pi^*}$ associated with an optimal policy $\pi^*$, which is unique if we assume that $\pi^*$ is unique. Likewise, let $d_t^b := d_t^{\pi^b}$ be the occupancy associated with $\pi^b$.

As in offline RL in MDPs, it is necessary to control the mismatch in occupancy between the behavior policy $\pi^b$ and the optimal policy $\pi^*$. Concretely, the single-policy RDP concentrability coefficient associated with RDP $\mathbf{R}$ and behavior policy $\pi^b$ is defined as in Cipollone et al. (2023):

$$C_{\mathbf{R}}^* = \max_{t \in [\![H]\!], u \in \mathcal{U}_t, ao \in \mathcal{AO}} \frac{d_t^*(u, ao)}{d_t^b(u, ao)} \,. \tag{2}$$

We assume that the concentrability is bounded away from infinity, i.e. that $C_{\mathbf{R}}^* < \infty$, which further implies that for any $t \in [\![H]\!]$, $u \in \mathcal{U}_t$, $d_t^b(u, a) > 0$ whenever $a \in \mathcal{A}$ is chosen by an optimal policy.

## 3 LEARNING RDPS WITH STATE-MERGING ALGORITHMS FROM OFFLINE DATA

To learn episodic RDPs from a dataset $\mathcal{D}$ of episodes, we adapt the algorithm ADACT-H (Cipollone et al., 2023). ADACT-H is a state-merging algorithm that iteratively constructs the set of RDP states $\mathcal{U}_0, \dots, \mathcal{U}_{H+1}$ and associated transition function $\tau$ of a minimal RDP $\mathbf{R}$. For each $t \in [\![0, H]\!]$, ADACT-H maintains a set of candidate states $\mathcal{U}_{c,t+1} = \{uao \mid u \in \mathcal{U}_t, ao \in \mathcal{AO}\}$ and a set of

promoted states $\mathcal{U}_{t+1}$. Each candidate state $uao$ has an associated multiset of suffixes $\mathcal{Z}(uao) = \{e^k_{t+1:H} : e^k \in \mathcal{D}, \bar{\tau}(e^k_{0:t-1}) = u, a^k_t o^k_t = ao\}$, i.e. episode suffixes whose history is consistent with $uao$. Initially, $\mathcal{U}_{t+1}$ contains the candidate with largest multiset. To decide if a candidate $uao$ should be promoted to $\mathcal{U}_{t+1}$ or merged with a promoted state $u' \in \mathcal{U}_{t+1}$, ADACT-H compares the empirical probability distributions on suffixes using the prefix distance $L^{\mathsf{p}}_\infty$. For reference, we include the pseudocode of ADACT-H$(\mathcal{D}, \delta)$ in Appendix A. Cipollone et al. (2023) prove that ADACT-H$(\mathcal{D}, \delta)$ constructs a minimal RDP $\mathbf{R}$ with probability at least $1 - 4AOU\delta$ if $\mathcal{D}$ is large enough.

In practice, the main bottleneck of ADACT-H is the statistical test on the last line,

$$L^{\mathsf{p}}_\infty(\mathcal{Z}_1, \mathcal{Z}_2) \geq \sqrt{2 \log(8(ARO)^{H-t}/\delta)/\min(|\mathcal{Z}_1|, |\mathcal{Z}_2|)},$$

where $\mathcal{Z}_1$ and $\mathcal{Z}_2$ are two multisets of traces in $\Gamma^{H-t}$ associated with a candidate state $uao$ and a promoted state $u'$, respectively. This is because the number of episode suffixes in $\Gamma^{H-t}$ is exponential in the current horizon $H - t$. The purpose of the present paper is to develop tractable methods for implementing the statistical test. These tractable methods can be directly applied to any algorithm that performs such statistical tests, e.g. the approximation algorithm ADACT-H-A (Cipollone et al., 2023) whose pseudocode we also include in Appendix A. Either algorithm can be incorporated into an offline RL algorithm for learning an $\varepsilon$-optimal policy, cf. Algorithm `RegORL` in Appendix A.

# 4 A TRACTABLE ALGORITHM FOR OFFLINE LEARNING OF RDPS

The lower bound derived in Cipollone et al. (2023) shows that sample complexity of learning RDPs is inversely proportional to the $L^{\mathsf{p}}_1$-distinguishability. When testing candidate states of the unknown RDP, $L^{\mathsf{p}}_1$ is the metric that allows maximum separation between distinct distributions over traces. Unfortunately, accurate estimates of $L^{\mathsf{p}}_1$ are impractical to obtain for distributions over large supports—in our case the number of episode suffixes which is exponential in the horizon. Accurate estimates of $L^{\mathsf{p}}_\infty$ are much more practical to obtain. However, there are instances for which states can be well separated by $L^{\mathsf{p}}_1$, but have an $L^{\mathsf{p}}_\infty$-distance that is exponentially small. To address these issues, in this section we develop two improvements over the previous learning algorithms for RDPs.

## 4.1 THE LANGUAGE METRIC

**Testing in the language metric** The metrics $L_1$ and $L_\infty$ are generic and can be applied to any distribution. Although this is generally an advantage, they do not exploit the internal structure of the sample space. In our application, a sample is a trace that, as discussed above, can be regarded as a string of a specific language. An important improvement, proposed by Balle (2013), is to consider $L^{\mathsf{p}}_1$ and $L^{\mathsf{p}}_\infty$, which take into account the variable length and conditional probabilities of longer suffixes. This was the approach followed by the previous RDP learning algorithms. However, these two metrics are substantially different and lead to dramatically different sample and space complexities.

In this section, we define a new formalism that unifies both metrics and will allow the development of new techniques for distinguishing distributions over traces. Specifically, instead of expressing the probability of single strings, we generalize the concepts above by considering the probability of *sets* of strings. A careful selection of the sets to consider, which are languages, will allow an accurate trade-off between generality and complexity.

**Definition 2** (Language metric). Let $\ell \in \mathbb{N}$, let $\Gamma$ be an alphabet, and let $\mathcal{X}$ be a set of languages consisting of strings in $\Gamma^\ell$. The *language metric*[1] in $\mathcal{X}$ is a function $L_\mathcal{X} : \Delta(\Gamma^\ell) \times \Delta(\Gamma^\ell) \to \mathbb{R}$, on pairs of probability distributions $p, p'$ over $\Gamma^\ell$, defined as

$$L_\mathcal{X}(p, p') := \max_{X \in \mathcal{X}} |p(X) - p'(X)|, \tag{3}$$

where the probability of a language is $p(X) := \sum_{x \in X} p(x)$.

This original notion unifies all the most common metrics. Considering distributions over $\Gamma^\ell$, when $\mathcal{X} = \{\{x\} \mid x \in \Gamma^\ell\}$, the language metric $L_\mathcal{X}$ reduces to $L_\infty$. When $\mathcal{X} = 2^{\Gamma^\ell}$, which is the set of all languages in $\Gamma^\ell$, the language metric becomes the total variation distance, which is half the value of $L_1$. A similar reduction can be made for the prefix distances. The language metric captures $L^{\mathsf{p}}_\infty$ when $\mathcal{X} = \{x\Gamma^{\ell-t} \mid t \in [\![0, \ell]\!], x \in \Gamma^t\}$, and it captures $L^{\mathsf{p}}_1$ when $\mathcal{X} = \cup_{t \in [\![0, \ell]\!]} 2^{\Gamma^t}$.

---

[1]$L_\mathcal{X}$ is only guaranteed to be a pseudometric, but we call it a metric for simplicity.

**Testing in language classes** The language metric can be applied directly to the language of traces and used for testing in RDP learning algorithms. The language of traces is over the alphabet $\Gamma = \mathcal{AO}/\mathcal{R}$. Thus, it suffices to consider any set of languages $\mathcal{X}$ that satisfy $X \subseteq \Gamma^{H-t} = (\mathcal{AO}/\mathcal{R})^{H-t}$ for each $X \in \mathcal{X}$. However, as we have seen above, the selection of a specific set of languages $\mathcal{X}$ has a dramatic impact on the metric that is being captured. In this section, we study an appropriate trade-off between generality and sample efficiency, obtained through a suitable selection of $\mathcal{X}$. Intuitively, we seek to evaluate the distance between candidate states based on increasingly complex sets of languages. We present a way to construct a hierarchy of sets of languages of increasing complexity. As a first step, we define sets of basic patterns $\mathcal{G}_i$ of increasing complexity.

$$\mathcal{G}_1 = \big\{a\mathcal{O}/\mathcal{R} \mid a \in \mathcal{A}\big\} \cup \big\{\mathcal{AO}/r \mid r \in \mathcal{R}\big\} \cup \big\{\mathcal{A}o/\mathcal{R} \mid o \in \mathcal{O}\big\},$$
$$\mathcal{G}_2 = \mathcal{G}_1 \cup \big\{ao/\mathcal{R} \mid a \in \mathcal{A}, o \in \mathcal{O}\big\} \cup \big\{a\mathcal{O}/r \mid a \in \mathcal{A}, r \in \mathcal{R}\big\} \cup \big\{\mathcal{A}o/r \mid a \in \mathcal{A}, r \in \mathcal{R}\big\},$$
$$\mathcal{G}_3 = \mathcal{G}_2 \cup \big\{ao/r \mid a \in \mathcal{A}, o \in \mathcal{O}, r \in \mathcal{R}\big\}.$$

The patterns $\mathcal{G}_1$ focus on single components, by matching a single action $a$, reward $r$, or observation $o$. Then, the patterns $\mathcal{G}_2$ focus on pairs, and the patterns $\mathcal{G}_3$ on triplets. Note that the number of basic patterns increases as we consider more interaction among actions, observations and rewards. For the first set we have $|\mathcal{G}_1| = A + O + R$, for the second $|\mathcal{G}_2| = |\mathcal{G}_1| + AO + AR + OR$, and for the third $|\mathcal{G}_3| = |\mathcal{G}_2| + AOR$. Starting from the above hierarchy, we identify one more dimension along which complexity can grow. It results from concatenating the basic patterns from $\mathcal{G}_i$, and is obtained by applying the operator $\mathrm{C}_k^\ell$ (Definition 1). Thus, letting $\ell = H - t$, we define the following *two-dimensional hierarchy* of sets $\mathcal{X}_{i,j}$ of languages. It is parameterised by $i$ for the granularity of the atomic symbols, and by $j$ for the sequential composition with operator $\mathrm{C}_k^\ell$. Formally,

$$\mathcal{X}_{i,j} = \bigcup_{k \in [\![j]\!]} \mathrm{C}_k^\ell(\mathcal{G}_i), \qquad \forall i \in [\![3]\!], \ \forall j \in [\![\ell]\!].$$

The family $\mathcal{X}_{i,j}$ induces a family of language metrics $L_{\mathcal{X}_{i,j}}$, which are non-decreasing along the dimensions of the hierarchy:

$$L_{\mathcal{X}_{i,j}} \le \min\big(L_{\mathcal{X}_{i+1,j}}, L_{\mathcal{X}_{i,j+1}}\big), \quad \forall i \in [\![3]\!], \forall j \in [\![\ell]\!].$$

The number of languages in each set grows as $|\mathcal{X}_{i,j}| \in \mathcal{O}(|\mathcal{G}_i|^j) = \mathcal{O}((AOR)^j)$, since the operator $\mathrm{C}_j^\ell$ introduces one language for each combination of $j$ languages from $\mathcal{G}_i$. Hence the family $L_{\mathcal{X}_{i,j}}$ has increasing complexity as we increase the values of the parameters $i$ and $j$. Moreover, the last level $\mathcal{X}_{3,\ell}$ satisfies $L_{\mathcal{X}_{3,\ell}} \ge L_\infty^{\mathrm{p}}$ since $\{x\mathcal{E}^{\ell-t} \mid t \in [\![0,\ell]\!], \ x \in \mathcal{E}^t\} \subseteq \mathcal{X}_{3,\ell}$. Therefore, the metric $L_{\mathcal{X}_{3,\ell}}$ is at least as effective as $L_\infty^{\mathrm{p}}$ in distinguishing states. It can be much more effective as shown next.

**Example 4** (Exponential gain in the T-maze). In the T-maze of Examples 1 and 3, when the behaviour policy $\pi^{\mathrm{b}}$ always chooses *East* in the corridor and *North* or *South* with equal probability at the T-junction, we observe the following exponential gap between the $L_\infty^{\mathrm{p}}$-distinguishability and the $L_{\mathcal{X}_{2,1}}$-distinguishability. The $L_\infty^{\mathrm{p}}$-distinguishability decreases exponentially with the corridor length $N$, since the distance between states is determined by the probability of single episode suffixes, which decreases exponentially with $N$ due to the random observations in the corridor. At the same time, the $L_{\mathcal{X}_{2,1}}$-distinguishability is constant. In this case, the distance between states is determined by the probability of observing a positive reward upon performing action *North*. This probability is $0.5$ in states of kind $\top$, and it is $0$ in states of kind $\bot$. More formally, the distance between states in $\mathcal{U}_t$ is determined by the probability of the language $\{x\, North\, \mathcal{O}/r\, y \mid x, y \in \Gamma^*, |xy| = H - t + 1\} \in \mathcal{X}_{2,1}$.

The argument of the example can be developed into a proof of the following theorem, showing that the language metric can yield an exponential improvement in distinguishability—proof in Appendix C.

**Theorem 1.** *There exist a family of RDPs $(\mathbf{R}_N)_{N \in \mathbb{N}}$ and a corresponding family of regular behaviour policies $(\pi_N^{\mathrm{b}})_{N \in \mathbb{N}}$ satisfying the two following properties: (i) the $L_\infty^{\mathrm{p}}$-distinguishability of $\mathbf{R}_N$ under $\pi_N^{\mathrm{b}}$ is $\mathcal{O}(2^{-N})$; and (ii) the $L_{\mathcal{X}_{2,1}}$-distinguishability of $\mathbf{R}_N$ under $\pi_N^{\mathrm{b}}$ is $\Omega(1)$.*

**Assumption 1.** The behavior policy $\pi^{\mathrm{b}}$ ensures an $L_{\mathcal{X}_{i,j}}$-distinguishability of at least $\mu_0 > 0$, where $\mathcal{X}_{i,j}$ is constructed as above and is an input to the algorithm.

Given $\ell \in [\![0, H]\!]$, let $p_1, p_2 \in \Delta(\Gamma^\ell)$ be two distributions over traces. To have accurate estimates for the language metric over some $\mathcal{X}_{i,j}$, we instantiate two estimators, $\widehat{p}_1$ and $\widehat{p}_2$, respectively built using multisets of traces $\mathcal{Z}_1$ and $\mathcal{Z}_2$, defined as the fraction of samples that belong to the language; that is, $\widehat{p}_1 := \sum_{e \in \mathcal{Z}_1} \mathbb{I}(e \in \mathcal{X}_{i,j})/|\mathcal{Z}_1|$ and $\widehat{p}_2 := \sum_{e \in \mathcal{Z}_2} \mathbb{I}(e \in \mathcal{X}_{i,j})/|\mathcal{Z}_2|$.

## 4.2 ANALYSIS

In this section we derive high-probability sample complexity bounds for ADACT-H (presented in Appendix A) when Count-Min-Sketch (CMS) or the language metric $L_\mathcal{X}$ is used.

**Theorem 2.** ADACT-H$(\mathcal{D}, \delta)$ *returns the minimal RDP* $\mathbf{R}$ *with probability at least* $1 - 3AOU\delta$ *when CMS is used to store empirical probability estimates, the statistical test is*

$$L_\infty^{\mathsf{p}}(\mathcal{Z}_1, \mathcal{Z}_2) \geq \sqrt{8\log(4(ARO)^{H-t}/\delta)/\min(|\mathcal{Z}_1|, |\mathcal{Z}_2|)},$$

*and the size of the dataset* $\mathcal{D}$ *is at least* $|\mathcal{D}| \geq \widetilde{\mathcal{O}}\left(\frac{HC_{\mathbf{R}}^* \log(1/\delta)}{d_m^* \cdot \mu_0^2}\right)$, *where* $d_m^* = \min_{t, u_t ao} d_t^*(u_t, ao)$ *is the minimum occupancy of the optimal policy* $\pi^*$.

The proof of Theorem 2 appears in Appendix C. Our new analysis uncovered a mistake in the proof of Cipollone et al. (2023), and as a result, both their and our sample complexity has an additional multiplicative term $\sqrt{H}/\mu_0$. The theorem shows that we can achieve the same asymptotic sample complexity when using CMS, but at a much lower memory cost.

**Theorem 3.** ADACT-H$(\mathcal{D}, \delta)$ *returns the minimal RDP* $\mathbf{R}$ *with probability at least* $1 - 2AOU\delta$ *when using the language metric* $L_\mathcal{X}$ *to define a statistical test*

$$L_\mathcal{X}(\mathcal{Z}_1, \mathcal{Z}_2) \geq \sqrt{2\log(2|\mathcal{X}|/\delta)/\min(|\mathcal{Z}_1|, |\mathcal{Z}_2|)},$$

*and the size of the dataset* $\mathcal{D}$ *is at least* $|\mathcal{D}| \geq \widetilde{\mathcal{O}}\left(\frac{C_{\mathbf{R}}^* \log(1/\delta)\log|\mathcal{X}|}{d_m^* \mu_0^2}\right)$.

The proof of Theorem 3 also appears in Appendix C. By definition, $\mu_0$ is the $L_\mathcal{X}$-distinguishability of $\mathbf{R}$ for the chosen language set $\mathcal{X}$, which has to satisfy $\mu_0 > 0$ for ADACT-H to successfully learn a minimal RDP. In the worst case, $\log|\mathcal{X}| = \widetilde{\mathcal{O}}(H)$, which matches the original bound. However, in the case of our language hierarchy $\mathcal{X}_{i,j}$, if $j$ is a small constant we have $\log|\mathcal{X}_{i,j}| = \mathcal{O}(\log((AOR)^j)) = \mathcal{O}(j\log(AOR)) = \widetilde{\mathcal{O}}(1)$. The constant $1/d_m^*$ depends exponentially on $H$ if there exists an RDP state that is very hard to reach, but can be much smaller for structured RDPs. In addition, as discussed earlier, $1/\mu_0$ may be exponentially smaller for $L_\mathcal{X}$ than for $L_\infty^{\mathsf{p}}$. For completeness, Appendix C also proves sample complexity bounds for CMS and $L_\mathcal{X}$ for the approximation algorithm ADACT-H-A.

## 5 EXPERIMENTAL RESULTS

In this section we present experimental results to illustrate the properties of our two versions of ADACT-H. We perform experiments in five domains from the literature on POMDPs and RDPs: Corridor (Ronca & De Giacomo, 2021), T-maze(c) (Bakker, 2001), Cookie (Toro Icarte et al., 2019), Cheese (McCallum, 1992) and Mini-hall (Littman et al., 1995), and summarize the results in Table 1. Appendix D contains a detailed description of each domain, as well as example automata learned. We compare against FlexFringe (Baumgartner & Verwer, 2023), a state-of-the-art algorithm for learning probabilistic-deterministic finite automata, which include RDPs as a special case. The RDPs output by FlexFringe are not always directly comparable to the RDPs output by ADACT-H, since FlexFringe sometimes learns RDPs with cycles, and uses a number of heuristics that optimize performance, but which no longer preserve the high-probability sample complexity guarantees.

| Name | $H$ | FlexFringe | | | CMS | | | Language metric | | |
|---|---|---|---|---|---|---|---|---|---|---|
| | | $U$ | $r$ | time | $U$ | $r$ | time | $U$ | $r$ | time |
| Corridor | 5 | 11 | **1.0** | 0.03 | 11 | **1.0** | 0.3 | 11 | **1.0** | 0.01 |
| T-maze(c) | 5 | 29 | 0.0 | 0.11 | 104 | **4.0** | 10.1 | 18 | **4.0** | 0.26 |
| Cookie | 9 | 220 | **1.0** | 0.36 | 116 | **1.0** | 6.05 | 91 | **1.0** | 0.08 |
| Cheese | 6 | 669 | $0.69 \pm .04$ | 19.28 | 1158 | $0.4 \pm .05$ | 207.4 | 1178 | $\mathbf{0.87 \pm .03}$ | 12.11 |
| Mini-hall | 15 | 897 | $0.33 \pm .04$ | 25.79 | - | - | - | 6098 | $\mathbf{0.86 \pm .03}$ | 29.90 |

Table 1: Summary of the experiments. For each domain, $H$ is the horizon, and for each algorithm, $U$ is the number of states of the learned automaton, $r$ is the reward of the derived policy, averaged over 100 episodes, and time is the running time in seconds of automaton learning. The maximum reward for all the domains is 1, except for T-maze(c) where the reward upon reaching the goal is 4.

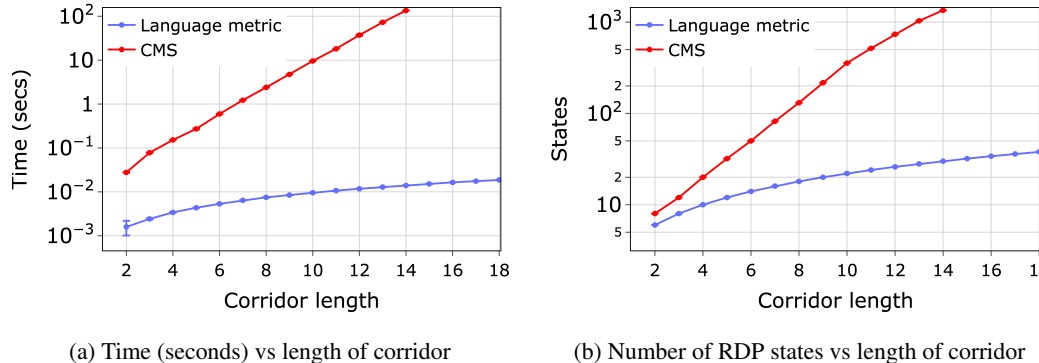

(a) Time (seconds) vs length of corridor      (b) Number of RDP states vs length of corridor

Figure 2: Impact of increasing the length of the corridor for the T-maze domain of Example 1.

In Table 1, we see that ADACT-H with the language family $\mathcal{X}_{3,1}$ is faster than FlexFringe in all domains except T-maze(c)—here FlexFringe fails to find the optimal policy, since the heuristics used are not optimized to preserve reward—and outputs smaller automata than both FlexFringe and CMS in all domains except Mini-hall. Since the number of languages in $\mathcal{X}_{3,1}$ is independent of $H$, the statistical test does not have to iterate over all suffixes, and the resulting RDPs better exploit the underlying structure of the domains. In the CMS-based approach, despite having a more compact representation of the probability distributions, the statistical test still has to iterate over all suffixes, which is exponential in $H$ for the $L^p_\infty$ distance, and exceeds the allotted time budget of 1800 seconds in the Mini-hall domain. In Corridor and Cookie, all algorithms learn automata that admit an optimal policy. However, in T-maze, Cheese and Mini-hall, the RDPs learned by ADACT-H with the language family $\mathcal{X}_{3,1}$ admit a policy that outperforms those of FlexFringe and CMS.

To better illustrate the improvement in performance for our language-based algorithm, in Figure 2 we compare the time and number of RDP states as a function of the corridor length in T-maze (Example 1) for our two methods based on CMS and the language metric, respectively. With the increasing corridor length $N$ (and horizon $H \simeq N$), we plot the time taken and RDP size over 20 runs, with $K = 100$ episodes. Figure 2a shows the time taken for the CMS-based algorithm increases exponentially whereas there is only a linear increase for the language-based approach, which is expected since the number of RDP states generated also increases linearly with $H$. In Figure 2b, the plot indicates the size of the RDP states produced by the language based approach is optimal ($\sim 2H$), and grows exponentially with $H$ for the CMS approach. We also extend this experiment by increasing $N$ up to 100, observing the same trend for our language-based approach. However, CMS suffers from slow running times, and exceeds 1800 seconds of running time over 20 runs after $H = 15$.

## 6   CONCLUSION

In this paper, we propose two new approaches to offline RL for Regular Decision Processes and provide their respective theoretical analysis. We also improve upon existing algorithms for RDP learning, and propose a modified algorithm using Count-Min-Sketch with reduced memory complexity. We define a hierarchy of language families and introduce a language-based approach, removing the dependency on $L^p_\infty$-distinguishability parameters and compare the performance of our algorithms to FlexFringe, a state-of-the-art algorithm for learning probabilistic deterministic finite automata. Although CMS suffers from a large running time, the language-restricted approach offers smaller automata and optimal (or near optimal) policies, even in domains requiring long-term dependencies. Finally, as a future work, we plan to expand our approach to the online RDP learning setting, and investigate how to learn RDPs with cycles.

## ACKNOWLEDGMENTS

This work has been co-funded by MCIN/AEI/10.13039/501100011033 under the Maria de Maeztu Units of Excellence Programme (CEX2021-001195-M). Anders Jonsson is partially supported by AGAUR 2021 SGR 00933 and Spanish grants PID2019-108141GB-I00 and UE-PID2023-147145NB-I00. Alessandro Ronca is supported by the European Research Council (ERC) under the European Union's Horizon 2020 research and innovation programme (Grant agreement No. 852769, ARiAT). Mohammad Sadegh Talebi acknowledges partial support by the Independent Research Fund Denmark, grant number 1026-00397B.

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

## A Pseudocode of AdaCT-H, AdaCT-H-A and RegORL

In this section we present the pseudocode of AdaCT-H, the approximation algorithm AdaCT-H-A and the full reinforcement learning algorithm RegORL.

---

**Function** AdaCT–H($\mathcal{D}$, $\delta$)

**Input:** Dataset $\mathcal{D}$ of traces in $\Gamma^{H+1}$, failure probability $0 < \delta < 1$
**Output:** Set $\mathcal{U}$ of RDP states, transition function $\tau : \mathcal{U} \times \mathcal{AO} \to \mathcal{U}$

1   $\mathcal{U}_0 \leftarrow \{u_0\}$, $\mathcal{Z}(u_0) \leftarrow \mathcal{D}$       // initial state
2   **for** $t = 0, \ldots, H$ **do**
3     $\mathcal{U}_{c,t+1} \leftarrow \{uao \mid u \in \mathcal{U}_t, ao \in \mathcal{AO}\}$      // get candidate states
4     **foreach** $uao \in \mathcal{U}_{c,t+1}$ **do** $\mathcal{Z}(uao) \leftarrow \{e_{t+1:H} \mid aroe_{t+1:H} \in \mathcal{Z}(u)\}$   // compute suffixes
5     $u_m a_m o_m \leftarrow \arg\max_{uao \in \mathcal{U}_{c,t+1}} |\mathcal{Z}(uao)|$      // most common candidate
6     $\mathcal{U}_{t+1} \leftarrow \{u_m a_m o_m\}$, $\tau(u_m, a_m o_m) = u_m a_m o_m$      // promote candidate
7     $\mathcal{U}_{c,t+1} \leftarrow \mathcal{U}_{c,t+1} \setminus \{u_m a_m o_m\}$      // remove from candidate states
8     **for** $uao \in \mathcal{U}_{c,t+1}$ **do**
9       $Similar \leftarrow \{u' \in \mathcal{U}_{t+1} \mid \text{not } \text{TESTDISTINCT}(t, \mathcal{Z}(uao), \mathcal{Z}(u'), \delta)\}$   // confidence test
10      **if** $Similar = \emptyset$ **then** $\mathcal{U}_{t+1} \leftarrow \mathcal{U}_{t+1} \cup \{uao\}$, $\tau(u, ao) = uao$     // promote candidate
11      **else** $u' \leftarrow$ element in $Similar$, $\tau(u, ao) = u'$, $\mathcal{Z}(u') \leftarrow \mathcal{Z}(u') \cup \mathcal{Z}(uao)$ // merge states
12     **end**
13 **end**
14 **return** $\mathcal{U}_0 \cup \cdots \cup \mathcal{U}_{H+1}$, $\tau$

15 **Function** TESTDISTINCT($t$, $\mathcal{Z}_1$, $\mathcal{Z}_2$, $\delta$)
16     **return** $L_\infty^p(\mathcal{Z}_1, \mathcal{Z}_2) \geq \sqrt{2\log(8(ARO)^{H-t}/\delta)/\min(|\mathcal{Z}_1|, |\mathcal{Z}_2|)}$

---

**Function** AdaCT–H–A($\mathcal{D}$, $\delta$, $\varepsilon$, $\overline{U}$, $\overline{C}$)

**Input:** Dataset $\mathcal{D}$ of traces in $\Gamma^{H+1}$, failure probability $0 < \delta < 1$, accuracy $\varepsilon \in (0, H]$, bounds $\overline{U}$ and $\overline{C}$
**Output:** Set $\mathcal{U}'$ of RDP states, transition function $\tau' : \mathcal{U}' \times \mathcal{AO} \to \mathcal{U}'$

1   $\mathcal{U}_0' \leftarrow \{u_0\}$, $\mathcal{Z}(u_0) \leftarrow \mathcal{D}$      // initial state
2   $\mathcal{U}_0' \leftarrow \mathcal{U}_0' \cup \{u_0^e\}$, $\mathcal{Z}(u_0^e) \leftarrow \emptyset$      // initial side state
3   **for** $t = 0, \ldots, H$ **do**
4     $\mathcal{U}_{t+1}' \leftarrow \{u_{t+1}^e\}$         // side state
5     **foreach** $ao \in \mathcal{AO}$ **do** $\tau'(u_t^e, ao) = u_{t+1}^e$, $\mathcal{Z}(u_{t+1}^e) \leftarrow \{e_{t+1:H} \mid aroe_{t+1:H} \in \mathcal{Z}(u_t^e)\}$
6     $\mathcal{U}_{c,t+1}' \leftarrow \{uao \mid u \in \mathcal{U}_t', ao \in \mathcal{AO}\}$      // get candidate states
7     **foreach** $uao \in \mathcal{U}_{c,t+1}'$ **do** $\mathcal{Z}(uao) \leftarrow \{e_{t+1:H} \mid aroe_{t+1:H} \in \mathcal{Z}(u)\}$   // compute suffixes
8     $u_m a_m o_m \leftarrow \arg\max_{uao \in \mathcal{U}_{c,t+1}'} |\mathcal{Z}(uao)|$      // most common candidate
9     $\mathcal{U}_{t+1}' \leftarrow \mathcal{U}_{t+1}' \cup \{u_m a_m o_m\}$, $\tau'(u_m, a_m o_m) = u_m a_m o_m$      // promote candidate
10    $\mathcal{U}_{c,t+1}' \leftarrow \mathcal{U}_{c,t+1}' \setminus \{u_m a_m o_m\}$      // remove from candidate states
11    **for** $uao \in \mathcal{U}_{c,t+1}'$ such that $|\mathcal{Z}(uao)|/N \geq 3\varepsilon/(10\overline{U}AO\overline{C})$ **do**
12      $Similar \leftarrow \{u' \in \mathcal{U}_{t+1}' \mid \text{not } \text{TESTDISTINCT}(t, \mathcal{Z}(uao), \mathcal{Z}(u'), \delta)\}$   // confidence test
13      **if** $Similar = \emptyset$ **then** $\mathcal{U}_{t+1}' \leftarrow \mathcal{U}_{t+1}' \cup \{uao\}$, $\tau'(u, ao) = uao$     // promote candidate
14      **else** $u' \leftarrow$ element in $Similar$, $\tau'(u, ao) = u'$, $\mathcal{Z}(u') \leftarrow \mathcal{Z}(u') \cup \mathcal{Z}(uao)$ // merge states
15      **if** $|\mathcal{U}_0'| + \cdots + |\mathcal{U}_{t+1}'| > \overline{U}$ **then return** Failure
16    **end**
17    **for** $uao \in \mathcal{U}_{c,t+1}'$ such that $|\mathcal{Z}(uao)|/N < 3\varepsilon/(10\overline{U}AO\overline{C})$ **do**
18      $\tau'(u, ao) = u_{t+1}^e$, $\mathcal{Z}(u_{t+1}^e) \leftarrow \mathcal{Z}(u_{t+1}^e) \cup \mathcal{Z}(uao)$      // merge with side state
19    **end**
20 **end**
21 **return** $\mathcal{U}_0' \cup \cdots \cup \mathcal{U}_{H+1}'$, $\tau'$

---

**Algorithm 1:** `RegORL`

---

**Input:** Dataset $\mathcal{D}$, accuracy $\varepsilon \in (0, H]$, failure probability $0 < \delta < 1$, (optionally) upper bound $\overline{U}$ on $|\mathcal{U}|$
**Output:** Policy $\hat{\pi} : \mathcal{H} \to \Delta(\mathcal{A})$

1   $\mathcal{D}_1, \mathcal{D}_2 \leftarrow$ separate $\mathcal{D}$ into two datasets of the same size
2   $\mathcal{U}, \tau \leftarrow$ ADACT–H$(\mathcal{D}_1, \delta/(4\overline{U}AO))$, where $\overline{U} = 2(AO)^H$ if not provided
3   $\mathcal{D}'_2 \leftarrow$ Markov transformation of $\mathcal{D}_2$ with respect to $\bar{\tau}$
4   $\hat{\pi}_{\mathsf{m}} \leftarrow$ OFFLINERL$(\mathcal{D}'_2, \varepsilon, \delta/2)$
5   **return** $\hat{\pi} : h \mapsto \hat{\pi}_{\mathsf{m}}(\bar{\tau}(h))$

---

## B   PROPERTIES OF REGULAR POLICIES

In this appendix we restate two propositions from Cipollone et al. (2023). We refer the reader to the proofs in that paper.

**Proposition 4.** *Consider an RDP* $\mathbf{R}$*, a regular policy* $\pi \in \Pi_{\mathbf{R}}$ *and two histories* $h_1$ *and* $h_2$ *in* $\mathcal{H}_t$*,* $t \in [H]$*, such that* $\bar{\tau}(h_1) = \bar{\tau}(h_2)$*. For each suffix* $e_{t+1:H} \in \Gamma^{H-t}$*, the probability of generating* $e_{t+1:H}$ *is the same for* $h_1$ *and* $h_2$*, i.e.* $\mathbb{P}(e_{t+1:H} \mid h_1, \pi, \mathbf{R}) = \mathbb{P}(e_{t+1:H} \mid h_2, \pi, \mathbf{R})$*.*

**Proposition 5.** *Each RDP* $\mathbf{R}$ *has at least one optimal policy* $\pi^* \in \Pi_{\mathbf{R}}$*.*

## C   PROOF OF THEOREMS

In this appendix we first prove Theorem 1. Then we prove Theorems 2 and 3 using a series of technical lemmas, and describing first how to implement ADACT-H using CMS.

### C.1   PROOF OF THEOREM 1: EXPONENTIAL IMPROVEMENT VIA THE LANGUAGE METRIC

**Theorem 1.** *There exist a family of RDPs* $(\mathbf{R}_N)_{N \in \mathbb{N}}$ *and a corresponding family of regular behaviour policies* $(\pi_N^{\mathsf{b}})_{N \in \mathbb{N}}$ *satisfying the two following properties: (i) the* $L_\infty^{\mathrm{p}}$*-distinguishability of* $\mathbf{R}_N$ *under* $\pi_N^{\mathsf{b}}$ *is* $\mathcal{O}(2^{-N})$*; and (ii) the* $L_{\mathcal{X}_{2,1}}$*-distinguishability of* $\mathbf{R}_N$ *under* $\pi_N^{\mathsf{b}}$ *is* $\Omega(1)$*.*

*Proof.* Let $\mathbf{R}_N$ be the RDP described in Example 3, where $N$ is the corridor length and the horizon is $H = N + 1$. Let $\pi_N^{\mathsf{b}}$ be the policy for $\mathbf{R}_N$ that always chooses *East* in the corridor, and chooses *North* or *South* with equal probability at the T-junction.

We show the first point by showing that the $L_\infty^{\mathrm{p}}$-distinguishability is at most $0.5^N$. It suffices to show

$$L_\infty^{\mathrm{p}}(\mathbb{P}(e_{1:H} \mid u_1 = u, \pi_N^{\mathsf{b}}), \mathbb{P}(e_{1:H} \mid u_1 = u', \pi_N^{\mathsf{b}})) \le 0.5^N,$$

for any two states $u, u' \in \mathcal{U}_1$. Then inequality is proven immediately since the $L_\infty^{\mathrm{p}}$-distance is upper-bounded by the probability of any single episode suffix $e_{1:H}$, which is $0.5^N$. In particular, a factor $0.5$ is due to the uniform choice of an action at the T-junction, and $0.5^{N-1}$ is due to the uniform probability over the two possible observations when we take a step in the corridor, for a total of $N - 1$ times.

We show the second point by showing that the $L_{\mathcal{X}_{2,1}}$-distinguishability is at least $0.5$. Let $t \in [\![N+1]\!]$, and let us consider two distinct states $u, u' \in \mathcal{U}_t$. It suffices to show

$$L_{\mathcal{X}_{2,1}}(\mathbb{P}(e_{t:H} \mid u_t = u, \pi_N^{\mathsf{b}}), \mathbb{P}(e_{t:H} \mid u_t = u', \pi_N^{\mathsf{b}})) \ge 0.5.$$

Since $\mathcal{U}_0 = \{u_0\}$ is a singleton, we have $t \ge 1$, and hence $u = (u_{t,\top}, t)$ and $u' = (u_{t,\perp}, t)$. The set of languages $\mathcal{X}_{2,1}$ includes $X = \{x \, North \, \mathcal{O}/r \, y \mid x, y \in \Gamma^*, |xy| = H - t + 1\}$, which is the language consisting of all episode suffixes where the positive reward $r$ is obtained upon performing action *North*. The probability of language $X$ is $0.5$ given $u_t = u$, and it is $0$ given $u_t = u'$. This proves the claimed inequality and hence the theorem. $\qquad\square$

## C.2 ANALYSIS OF CMS

Given a finite set $\mathcal{Y} = \{y_1, \ldots, y_m\}$ and a probability distribution $p \in \Delta(\mathcal{Y})$, let $\widehat{p} \in \Delta(\mathcal{Y})$ be an empirical estimate of $p$ computed using $n$ samples. CMS can store an estimate $\widetilde{p}$ of the empirical distribution $\widehat{p}$. In this setting, the vector $v = (v_1, \ldots, v_m)$ contains the empirical counts of each element of $\mathcal{Y}$, i.e. for each $i \in [\![m]\!]$, $v_i$ is the number of times that we have observed element $y_i$. Given $n$ samples, we have $\|v\|_1 = n$ and $\widehat{p}(y_i) = v_i/n$ for each $y_i \in \mathcal{Y}$. The following lemma shows how to bound the error between $\widetilde{p}$ and $\widehat{p}$.

**Lemma 6.** *Given a finite set $\mathcal{Y}$, a probability distribution $p \in \Delta(\mathcal{Y})$, and an empirical estimate $\widehat{p} \in \Delta(\mathcal{Y})$ of $p$ obtained using $n$ samples, let $\widetilde{p}$ be the estimate of $\widehat{p}$ output by CMS with parameters $\delta_c$ and $\varepsilon$. With probability at least $1 - |\mathcal{Y}|\delta_c$ it holds that $\|\widetilde{p} - \widehat{p}\|_\infty \leq \varepsilon$.*

*Proof.* Cormode & Muthukrishnan (2005) show that for a point query that returns an approximation $\widetilde{v}_i$ of $v_i$, with probability at least $1 - \delta_c$ it holds that

$$\widetilde{v}_i \leq v_i + \varepsilon\|v\|_1.$$

In our case, the estimated probability of an element $y_i \in \mathcal{Y}$ equals $\widetilde{p}(y_i) = \widetilde{v}_i/n$, where $\widetilde{v}_i$ is the point query for $y_i$. Using the result above, with probability at least $1 - \delta_c$ we have

$$\widetilde{p}(y_i) = \frac{\widetilde{v}_i}{n} \leq \frac{v_i}{n} + \frac{\varepsilon\|v\|_1}{n} = \widehat{p}(y_i) + \varepsilon.$$

Since CMS never underestimates a value, $\widehat{p}(y_i) \leq \widetilde{p}(y_i)$ trivially holds. Taking a union bound shows that the inequality above holds simultaneously for all $y_i \in \mathcal{Y}$ with probability $1 - |\mathcal{Y}|\delta_c$. $\square$

In our setting, $\mathcal{Y} = \Gamma^\ell$ for a fixed $\ell \in [\![H]\!]$. Given a failure probability $\delta$ and a multiset $\mathcal{Z}$ of elements in $\Gamma^\ell$, we set the parameters of CMS to $\delta_c = \delta/(AOR)^\ell$ and $\varepsilon = \sqrt{\log(2(AOR)^\ell/\delta)/2|\mathcal{Z}|}$. This implies that the size of the matrix of CMS is proportional to

$$\log\left(\frac{1}{\delta_c}\right) \cdot \frac{e}{\varepsilon} = e \log\left(\frac{(AOR)^\ell}{\delta}\right) \sqrt{\frac{2|\mathcal{Z}|}{\log\left(2(AOR)^\ell/\delta\right)}} < e\sqrt{2|\mathcal{Z}| \log\left(\frac{2(AOR)^\ell}{\delta}\right)}$$

$$= \widetilde{\mathcal{O}}\left(\sqrt{|\mathcal{Z}|(\ell + \log(1/\delta))}\right).$$

Hence the memory complexity is much smaller than $|\Gamma^\ell|$, the number of elements in $v$. Since $(AOR)^\ell = |\Gamma^\ell|$, Lemma 6 implies that $\|\widetilde{p} - \widehat{p}\|_\infty \leq \epsilon$ holds with probability at least $1 - \delta$.

Given a language $X \subseteq \Gamma^\ell$, the estimation error $|\widetilde{p}(X) - \widehat{p}(X)|$ due to CMS is proportional to $|X|\varepsilon$, which is large if $X$ contains many elements. Alternatively, for a language set $\mathcal{X} = \{X_1, \ldots, X_m\}$, we could use the vector element $v_i$ of CMS to count the number of elements of $\mathcal{Z}$ that are also members of $X_i$. However, if the languages overlap (which is the case for many of our language families), then $\|v\|_1 = n$ no longer holds, and neither does Lemma 6. Hence it is only practical to use CMS to estimate the error of the language metric $L_\mathcal{X}$ when the languages in $\mathcal{X}$ are small and disjoint.

To bound the prefix distance $L_\infty^p(\widetilde{p}_{uao}, \widehat{p}_{uao})$ between the empirical and estimated distributions on suffixes of length $H - t$, for each candidate state $(u, ao) \in \mathcal{U}_t \times \mathcal{AO}$ we need $H - t$ copies of CMS to store the counts for suffixes of different lengths $\ell \in [\![H - t]\!]$. We define an associated event $\mathcal{E}_{\text{CMS}}$ to correctly bound $L_\infty^p(\widetilde{p}_{uao}, \widehat{p}_{uao})$ for all candidate states:

$$\mathcal{E}_{\text{CMS}} = \left\{\forall t \in [\![0, H]\!], (u, ao) \in \mathcal{U}_t \times \mathcal{AO} : L_\infty^p(\widetilde{p}_{uao}, \widehat{p}_{uao}) \leq \varepsilon = \sqrt{\frac{\log(2(AOR)^{H-t}/\delta)}{2|\mathcal{Z}|}}\right\}.$$

Since the number of instances of CMS is less than $HUAO$, event $\mathcal{E}_{\text{CMS}}$ occurs with probability at least $1 - HUAO\delta$.

## C.3 TECHNICAL LEMMAS

The following technical lemmas largely follow the analysis of Cipollone et al. (2023). However, we reformulate the lemmas in terms of the language metric $L_\mathcal{X}$ and make them more modular in order to adapt them to our different settings. We first prove a high-probability upper bound on the language metric $L_\mathcal{X}$ between the true suffix distribution and its empirical estimate for any candidate state.

**Lemma 7.** *Given a timestep $t \in [\![0, H]\!]$, a candidate state $(u, ao) \in \mathcal{U}_t \times \mathcal{AO}$, a multiset $\mathcal{Z}(uao)$ of suffixes in $\Gamma^{H-t}$, and a set $\mathcal{X}$ of languages defined on $\Gamma^{H-t}$, with probability at least $1 - \delta$ the language metric $L_{\mathcal{X}}$ satisfies*

$$L_{\mathcal{X}}(\widehat{p}_{uao}, p_{uao}) \leq \sqrt{\frac{\log(2|\mathcal{X}|/\delta)}{2|\mathcal{Z}(uao)|}},$$

*where $p_{uao} \in \Delta(\Gamma^{H-t})$ is the true distribution on suffixes in $\Gamma^{H-t}$ that start from $(u, ao)$, and $\widehat{p}_{uao}$ is the empirical estimate of $p_{uao}$ induced by $\mathcal{Z}(uao)$.*

*Proof.* For each language $X \in \mathcal{X}$, let $p_{uao}(X) = \sum_{x \in X} p_{uao}(x)$ be the true probability of $X$, and let $\widehat{p}_{uao}(X) = \sum_{x \in \mathcal{Z}(uao)} \mathbb{I}(x \in X)/|\mathcal{Z}(uao)|$ be the empirical estimate of $p_{uao}(X)$, i.e. the proportion of elements in $\mathcal{Z}(uao)$ that are also in $X$. Hoeffding's inequality implies that

$$\mathbb{P}\left( |\widehat{p}_{uao}(X) - p_{uao}(X)| > \sqrt{\frac{\log(2/\delta_{\mathsf{s}})}{2|\mathcal{Z}(uao)|}} \right) \leq \delta_{\mathsf{s}}.$$

Choosing $\delta_{\mathsf{s}} = \delta/|\mathcal{X}|$ and taking a union bound implies that $L_{\mathcal{X}}$ satisfies

$$L_{\mathcal{X}}(\widehat{p}_{uao}, p_{uao}) = \max_{X \in \mathcal{X}} |\widehat{p}_{uao}(X) - p_{uao}(X)| \leq \sqrt{\frac{\log(2|\mathcal{X}|/\delta)}{2|\mathcal{Z}(uao)|}}$$

with probability $1 - |\mathcal{X}|\delta_{\mathsf{s}} = 1 - \delta$, which completes the proof. $\qquad\square$

We define an associated event $\mathcal{E}_{\mathcal{X}}$ to correctly bound the language metric $L_{\mathcal{X}}$ for all $(u, ao)$:

$$\mathcal{E}_{\mathcal{X}} = \left\{ \forall t \in [\![0, H]\!], (u, ao) \in \mathcal{U}_t \times \mathcal{AO} : L_{\mathcal{X}}(\widehat{p}_{uao}, p_{uao}) \leq \sqrt{\frac{\log(2|\mathcal{X}|/\delta)}{2|\mathcal{Z}(uao)|}} \right\}.$$

We next prove a high-probability sample complexity bound for accurately estimating the empirical occupancy $\widehat{p}(uao)$ of each candidate state $uao \in \mathcal{U} \times \mathcal{AO}$. Given a number of episodes $N$, an empirical Bernstein inequality yields

$$\mathbb{P}\left( |\widehat{p}(uao) - d_t^{\mathsf{b}}(u, ao)| > \sqrt{\frac{2\widehat{p}(uao)\log(4/\delta)}{N}} + \frac{14\log(4/\delta)}{3N} \right) \leq \delta. \qquad (4)$$

Given a failure probability $\delta$, let $G_\delta$ be the function for the bound in the empirical Bernstein inequality, defined as

$$G_\delta(\widehat{p}, N) = \sqrt{\frac{2\widehat{p}\log(4/\delta)}{N}} + \frac{14\log(4/\delta)}{3N}$$

It is easy to see that $G_\delta$ is monotonically increasing in $\widehat{p}$ and monotonically decreasing in $N$. We define an associated event $\mathcal{E}_B$ to correctly bound $|\widehat{p}(uao) - d_t^{\mathsf{b}}(u, ao)|$ for all $(u, ao)$:

$$\mathcal{E}_B = \left\{ \forall t \in [\![0, H]\!], (u, ao) \in \mathcal{U}_t \times \mathcal{AO} : |\widehat{p}(uao) - d_t^{\mathsf{b}}(u, ao)| \leq G_\delta(\widehat{p}(uao), N) \right\}.$$

We show that we can control the number of episodes $N$ to obtain an upper bound on the function $G_\delta$.

**Lemma 8.** *For fixed probabilities $\delta$ and $\widehat{p}$, if $N \geq 16\log(4/\delta)/\widehat{p}$ it holds that $3G_\delta(\widehat{p}, N) < 2\widehat{p}$.*

*Proof.* We first show that the inequality holds for $N = 16\log(4/\delta)/\widehat{p}$. In this case we have

$$3G_\delta(\widehat{p}, N) = 3\sqrt{\frac{2\widehat{p}^2\log(4/\delta)}{16\log(4/\delta)} + \frac{14\widehat{p}\log(4/\delta)}{16\log(4/\delta)}} = \left( \frac{3}{\sqrt{8}} + \frac{14}{16} \right)\widehat{p} < 2\widehat{p}.$$

The case $N > 16\log(4/\delta)/\widehat{p}$ follows from the fact that $G_\delta$ is monotonically decreasing in $N$. $\qquad\square$

Since $\widehat{p}(uao) = |\mathcal{Z}(uao)|/N$ implies $N = |\mathcal{Z}(uao)|/\widehat{p}(uao)$, we obtain the following corollary.

**Corollary 9.** *Under event $\mathcal{E}_B$, if $|\mathcal{Z}(uao)| \geq 16 \log(4/\delta)$ it holds that $|\widehat{p}(uao) - d_t^{\mathsf{b}}(u, ao)| \leq 2\widehat{p}(uao)/3$.*

We show that under event $\mathcal{E}_B$, we can choose the sample complexity $N$ to ensure that we obtain at least a certain number of elements in $\mathcal{Z}(uao)$.

**Lemma 10.** *Given a candidate state $uao \in \mathcal{U}_t \times \mathcal{AO}$ and any $b \geq 1$, under event $\mathcal{E}_B$ it holds that $|\mathcal{Z}(uao)| \geq b \log(4/\delta)$ if the sample complexity $N$ satisfies*

$$N \geq \frac{\log(4/\delta)}{d_t^{\mathsf{b}}(u, ao)} \left(2b + 31/6\right).$$

*Proof.* Letting $M = |\mathcal{Z}(uao)|$, due to event $\mathcal{E}_B$ and the given bound on $N$ it holds that

$$
\begin{aligned}
d_t^{\mathsf{b}}(u, ao) - \frac{M}{N} &\leq G_\delta(M/N, N) \\
\Leftrightarrow \quad 0 &\leq M + N G_\delta(M/N, N) - N d_t^{\mathsf{b}}(u, ao) \\
&\leq M + \sqrt{2M \log(4/\delta)} + 14 \log(4/\delta)/3 - \log(4/\delta)\left(2b + 31/6\right) \\
&= M + \sqrt{2 \log(4/\delta)}\sqrt{M} - \log(4/\delta)\left(2b + 1/2\right).
\end{aligned}
$$

Solving the quadratic inequality for positive $\sqrt{M}$ yields

$$
\begin{aligned}
\sqrt{M} &\geq -\sqrt{\frac{\log(4/\delta)}{2}} + \sqrt{\frac{\log(4/\delta)}{2} + \log(4/\delta)\left(2b + 1/2\right)} \\
&= -\sqrt{\frac{\log(4/\delta)}{2}} + \sqrt{\log(4/\delta) + 2b \log(4/\delta)} \\
&\geq -\sqrt{\frac{\log(4/\delta)}{2}} + \frac{\sqrt{\log(4/\delta)} + \sqrt{2b \log(4/\delta)}}{\sqrt{2}} = \sqrt{b \log(4/\delta)},
\end{aligned}
$$

where we have used the inequality $\sqrt{x + y} \geq (\sqrt{x} + \sqrt{y})/\sqrt{2}$. Hence the bound on $N$ in the lemma implies that $M = \sqrt{M}^2 \geq b \log(4/\delta)$. $\qquad\square$

### C.4 PROOF OF THEOREM 2

Let $\text{TESTDISTINCT}_{CMS}$ be the version of $\text{TESTDISTINCT}$ that uses the statistical test in the theorem. To prove Theorem 2, we show that $\text{TESTDISTINCT}_{CMS}$ answers correctly if the multisets $\mathcal{Z}_1$ and $\mathcal{Z}_2$ have a given minimum size $M_{CMS}$, both when the suffix distributions are the same and when they are different. These lemmas are analogous to Lemmas 13 and 14 of Cipollone et al. (2023).

**Lemma 11.** *For $t \in [\![0, H]\!]$, let $\mathcal{Z}_1$ and $\mathcal{Z}_2$ be multisets sampled from distributions $p_1$ and $p_2$ on $\Delta(\Gamma^{H-t})$. If we use CMS with parameters $\varepsilon = \sqrt{\log(2(AOR)^{H-t}/\delta)/2|\mathcal{Z}|}$ and $\delta_c = \delta/(AOR)^{H-t}$ to store an approximation $\widetilde{p}_i$ of the empirical estimate $\widehat{p}_i$ of $p_i$ induced by $\mathcal{Z}_i$, $i \in [\![2]\!]$, if events $\mathcal{E}_{CMS}$ and $\mathcal{E}_{\mathcal{X}}$ hold, and if $p_1 = p_2$, then $\text{TESTDISTINCT}_{CMS}(t, \mathcal{Z}_1, \mathcal{Z}_2, \delta)$ returns false.*

*Proof.* For $i \in [\![2]\!]$, $L_\infty^{\mathsf{p}}(\widehat{p}_i, p_i) = L_{\mathcal{X}}(\widehat{p}_i, p_i)$ if $\mathcal{X}$ contains one language for each prefix from $\Gamma^{H-t}$, implying $|\mathcal{X}| \leq 2(AOR)^{H-t}$. Events $\mathcal{E}_{CMS}$ and $\mathcal{E}_{\mathcal{X}}$ and the triangle inequality now imply that

$$
\begin{aligned}
L_\infty^{\mathsf{p}}(\widetilde{p}_1, \widetilde{p}_2) &\leq L_\infty^{\mathsf{p}}(\widetilde{p}_1, \widehat{p}_1) + L_{\mathcal{X}}(\widehat{p}_1, p_1) + L_{\mathcal{X}}(p_1, p_2) + L_{\mathcal{X}}(p_2, \widehat{p}_2) + L_\infty^{\mathsf{p}}(\widehat{p}_2, \widetilde{p}_2) \\
&\leq \sqrt{\frac{\log(2(AOR)^{H-t}/\delta)}{2|\mathcal{Z}_1|}} + \sqrt{\frac{\log(2|\mathcal{X}|/\delta)}{2|\mathcal{Z}_1|}} + 0 + \sqrt{\frac{\log(2|\mathcal{X}|/\delta)}{2|\mathcal{Z}_2|}} + \sqrt{\frac{\log(2(AOR)^{H-t}/\delta)}{2|\mathcal{Z}_2|}} \\
&\leq \sqrt{\frac{8 \log(4(AOR)^{H-t}/\delta)}{\min(|\mathcal{Z}_1|, |\mathcal{Z}_2|)}}.
\end{aligned}
$$

This is precisely the condition for which $\text{TESTDISTINCT}_{CMS}$ returns false. $\qquad\square$

**Lemma 12.** *For $t \in [\![0, H]\!]$, let $\mathcal{Z}_1$ and $\mathcal{Z}_2$ be multisets sampled from distributions $p_1$ and $p_2$ on $\Delta(\Gamma^{H-t})$. If we use CMS with parameters $\varepsilon = \sqrt{\log(2(AOR)^{H-t}/\delta)/2|\mathcal{Z}|}$ and $\delta_c = \delta/(AOR)^{H-t}$ to store an approximation $\widetilde{p}_i$ of the empirical estimate $\widehat{p}_i$ of $p_i$ induced by $\mathcal{Z}_i$, $i \in [\![2]\!]$, if events $\mathcal{E}_{CMS}$ and $\mathcal{E}_\mathcal{X}$ hold, and if $p_1 \neq p_2$, then $\text{TESTDISTINCT}_{CMS}(t, \mathcal{Z}_1, \mathcal{Z}_2, \delta)$ returns true if $\mathcal{Z}_1$ and $\mathcal{Z}_2$ satisfy $\min(|\mathcal{Z}_1|, |\mathcal{Z}_2|) \geq 32 \log(4(AOR)^{H-t}/\delta)/\mu_0^2$.*

*Proof.* Using the same argument as in the proof of Lemma 11 yields

$$L_\infty^{\mathsf{p}}(\widetilde{p}_1, \widetilde{p}_2) \geq L_\mathcal{X}(p_1, p_2) - L_\infty^{\mathsf{p}}(\widetilde{p}_1, \widehat{p}_1) - L_\mathcal{X}(\widehat{p}_1, p_1) - L_\mathcal{X}(p_2, \widehat{p}_2) - L_\infty^{\mathsf{p}}(\widehat{p}_2, \widetilde{p}_2)$$

$$\geq \mu_0 - \sqrt{\frac{\log(2(AOR)^{H-t}/\delta)}{2|\mathcal{Z}_1|}} - \sqrt{\frac{\log(2|\mathcal{X}|/\delta)}{2|\mathcal{Z}_1|}} - \sqrt{\frac{\log(2|\mathcal{X}|/\delta)}{2|\mathcal{Z}_2|}} - \sqrt{\frac{\log(2(AOR)^{H-t}/\delta)}{2|\mathcal{Z}_2|}}$$

$$\geq \mu_0 - \sqrt{\frac{8 \log(4(AOR)^{H-t}/\delta)}{\min(|\mathcal{Z}_1|, |\mathcal{Z}_2|)}} \geq \mu_0 - \sqrt{\frac{\mu_0^2}{4}} = \frac{\mu_0}{2} \geq \sqrt{\frac{8 \log(4(AOR)^{H-t}/\delta)}{\min(|\mathcal{Z}_1|, |\mathcal{Z}_2|)}},$$

where we have used the condition on $\min(|\mathcal{Z}_1|, |\mathcal{Z}_2|)$ in the lemma. This is precisely the condition for which $\text{TESTDISTINCT}_{CMS}$ returns true. $\quad\square$

We next show that ADACT-H returns a minimal RDP if the multiset $\mathcal{Z}(uao)$ associated with each candidate state $uao$ satisfies $|\mathcal{Z}(uao)| \geq 32 \log(4(AOR)^{H-t}/\delta)/\mu_0^2 \equiv M_{CMS}$.

**Lemma 13.** *Under events $\mathcal{E}_{CMS}$ and $\mathcal{E}_\mathcal{X}$, ADACT-H outputs a minimal RDP $\mathbf{R}$ if the multiset $\mathcal{Z}(uao)$ associated with each candidate state $uao$ satisfies $|\mathcal{Z}(uao)| \geq M_{CMS}$.*

*Proof.* We show the result by induction. The base case is given by the set $\mathcal{U}_0$, which is clearly minimal since it only contains the initial state $u_0$. For $t \in [\![0, H]\!]$, assume that the algorithm has learned a minimal RDP for sets $\mathcal{U}_0, \ldots, \mathcal{U}_t$. Let $\mathcal{U}_{t+1}$ be the set of states at layer $t + 1$ of a minimal RDP. Due to the regular property, each pair of histories that map to a state $u_{t+1} \in \mathcal{U}_{t+1}$ generate the same probability distribution over suffixes. Under events $\mathcal{E}_{CMS}$ and $\mathcal{E}_\mathcal{X}$, Lemma 11 implies that $\text{TESTDISTINCT}_{CMS}(t, \mathcal{Z}(uao), \mathcal{Z}(u'a'o'), \delta)$ returns false for each pair of candidate states $uao$ and $u'a'o'$ that map to $u_{t+1}$. Consequently, ADACT-H merges $uao$ and $u'a'o'$.

On the other hand, by assumption, each pair of histories that map to different states of $\mathcal{U}_{t+1}$ have $L_\infty$-distinguishability $\mu_0$. Under events $\mathcal{E}_{CMS}$ and $\mathcal{E}_\mathcal{X}$, if $\min(|\mathcal{Z}(uao)|, |\mathcal{Z}(u'a'o')|) \geq M_{CMS}$ then Lemma 12 implies that $\text{TESTDISTINCT}_{CMS}(t, \mathcal{Z}(uao), \mathcal{Z}(u'a'o'), \delta)$ returns true for each pair of candidate states $uao$ and $u'a'o'$ that map to different states in $\mathcal{U}_{t+1}$. Consequently, ADACT-H does not merge $uao$ and $u'a'o'$. It follows that ADACT-H will generate exactly the set $\mathcal{U}_{t+1}$, which is that of a minimal RDP. $\quad\square$

To prove Theorem 2 we need to ensure that $M \geq M_{CMS}$ for each candidate state $uao$. Choosing $b = 64 \log(AOR)^H/\mu_0^2$ and applying Lemma 10 yields the following bound on the sample complexity:

$$N \geq \max_{uao} \left\{ \frac{\log(4/\delta)}{d_t^{\mathsf{b}}(u, ao)} \left( \frac{128 \log(AOR)^H}{\mu_0^2} + 31/6 \right) \right\}.$$

The three events $\mathcal{E}_{CMS}$, $\mathcal{E}_\mathcal{X}$ and $\mathcal{E}_B$ hold simultaneously with probability at least $1 - (H + 2)UAO\delta \geq 1 - 2HUAO\delta$, so defining $\delta_0 = \delta/2HUAO$ ensures that the sample complexity holds with probability at least $1 - \delta_0$. Since $d_t^{\mathsf{b}}(u, ao) \geq d_t^*(u, ao)/C_{\mathbf{R}}^* \geq d_{\mathsf{m}}^*/C_{\mathbf{R}}^*$, we can bound the sample complexity as

$$N \geq \frac{C_{\mathbf{R}}^* \log(8HUAO/\delta_0)}{d_{\mathsf{m}}^*} \left( \frac{128 \log(AOR)^H}{\mu_0^2} + 31/6 \right) = \widetilde{\mathcal{O}} \left( \frac{HC_{\mathbf{R}} \log(1/\delta)}{d_{\mathsf{m}}^* \cdot \mu_0^2} \right).$$

This concludes the proof of the theorem.

We also prove a bound on the sample complexity of the approximation algorithm ADACT-H-A. In this algorithm, the subroutine $\text{TESTDISTINCT}_{CMS}$ is only called for a candidate state $uao$ when $\widehat{p}(uao)$ satisfies

$$\widehat{p}(uao) \geq \frac{3\varepsilon}{10\overline{U}\,AO\overline{C}} \equiv \psi,$$

where $\varepsilon$, $\overline{U}$ and $\overline{C}$ are inputs to the algorithm and $\psi$ is a predefined threshold. We prove the following theorem:

**Theorem 14.** ADACT-H-A$(\mathcal{D}, \delta, \varepsilon, \overline{U}, \overline{C})$ *returns an $\frac{\varepsilon}{2}$-approximate RDP $\mathbf{R}'$ with probability at least $1 - 2HAOU\delta$ when CMS is used to store empirical probability estimates, the statistical test is*

$$L_\infty^{\mathsf{p}}(\mathcal{Z}_1, \mathcal{Z}_2) \geq \sqrt{8\log(4(ARO)^{H-t}/\delta)/\min(|\mathcal{Z}_1|, |\mathcal{Z}_2|)},$$

*and the size of the dataset $\mathcal{D}$ is at least*

$$|\mathcal{D}| \geq \widetilde{\mathcal{O}}\left(\frac{H\overline{U}AO\overline{C}\log(1/\delta)}{\varepsilon\mu_0^2}\right).$$

We first prove that the resulting RDP is $\frac{\varepsilon}{2}$-approximate.

**Lemma 15.** *Under events $\mathcal{E}_{CMS}$, $\mathcal{E}_\mathcal{X}$ and $\mathcal{E}_B$, if $\overline{U}$ and $\overline{C}$ are upper bounds on the number of RDP states $U' = |\mathcal{U}'|$ and concentrability $C_{\mathbf{R}'}^*$, then* ADACT-H-A *returns an $\frac{\varepsilon}{2}$-approximate RDP $\mathbf{R}'$.*

*Proof.* Consider a candidate state $uao$ with $M = |\mathcal{Z}(uao)|$. If $\widehat{p}(uao) \geq \psi$ then we impose the condition $M \geq M_{CMS}$ as before. For each such candidate state, ADACT-H-A calls TESTDIS-TINCT$_{CMS}$ and correctly promotes the candidate state to an RDP state or merges it with an existing RDP state.

On the other hand, if $\widehat{p}(uao) < \psi$ and $N \geq 16\log(4/\delta)/\psi$, event $\mathcal{E}_B$ and Lemma 8 yield

$$d_t^{\mathsf{b}}(u, ao) - \widehat{p}(uao) \leq G_\delta(\widehat{p}(qao), N)$$

$$\Leftrightarrow \quad d_t^{\mathsf{b}}(u, ao) < \widehat{p}(uao) + G_\delta(\widehat{p}(uao), N) < \psi + G_\delta(\psi, N) \leq \frac{5\psi}{3} = \frac{\varepsilon}{2\overline{U}AO\overline{C}}.$$

In this case, ADACT-H-A does not call TESTDISTINCT$_{CMS}$ and hence the resulting RDP state may be incorrect. We can bound the contribution of $uao$ to the value under the optimal policy $\pi^*$ as

$$d_t^*(u, ao) \sum_{a' \in \mathcal{A}} \pi^*(\tau(u, ao), a') \sum_{r \in \mathcal{R}} \theta_{\mathsf{r}}(\tau(u, ao), a', r) \cdot r$$

$$\leq d_t^*(u, ao) \sum_{a' \in \mathcal{A}} \pi^*(\tau(u, ao), a') \sum_{r \in \mathcal{R}} \theta_{\mathsf{r}}(\tau(u, ao), a', r) = d_t^*(u, ao) \leq C_{\mathbf{R}'}^* d_t^{\mathsf{b}} \leq \frac{\varepsilon}{2\overline{U}AO},$$

where we have used the fact that the reward is bounded by 1. Summing up the contribution of all such candidate states to the expected optimal value of histories in $\mathcal{H}_0$ yields

$$\sum_{t \in [\![0,H]\!]} \sum_{u_t ao} d_t^*(u_t, ao) \sum_{a' \in \mathcal{A}} \pi^*(\tau(u, ao), a') \sum_{r \in \mathcal{R}} \theta_{\mathsf{r}}(\tau(u, ao), a', r) \cdot r \leq \sum_{t \in [\![0,H]\!]} \sum_{u_t ao} \frac{\varepsilon}{2\overline{U}AO} \leq \frac{\varepsilon}{2}.$$

This proves that the resulting RDP $\mathbf{R}'$ is $\frac{\varepsilon}{2}$-approximate. $\qquad\square$

To prove Theorem 14, for each candidate state $uao$ such that $\widehat{p}(uao) < \psi$, a number of episodes which satisfies $N \geq 16\log(4/\delta)/\psi$ is sufficient to ensure that $\mathbf{R}'$ is $\frac{\varepsilon}{2}$-approximate. If $\widehat{p}(uao) \geq \psi$, we instead require $M \geq M_{CMS}$ as before. Since $M_{CMS} \geq 16\log(4/\delta)$, event $\mathcal{E}_B$ together with Corollary 9 yield

$$\widehat{p}(uao) - d_t^{\mathsf{b}}(u, ao) \leq \frac{2\widehat{p}(uao)}{3} \quad \Leftrightarrow \quad d_t^{\mathsf{b}}(u, ao) \geq \frac{\widehat{p}(uao)}{3} \geq \frac{\psi}{3} = \frac{\varepsilon}{10\overline{U}AO\overline{C}}.$$

Applying Lemma 10 directly yields the following bound on the sample complexity:

$$N \geq \max_{uao}\left\{\frac{\log(4/\delta)}{d_t^{\mathsf{b}}(u, ao)}\left(\frac{128\log(AOR)^H}{\mu_0^2} + 31/6\right) + \frac{16\log(4/\delta)}{\psi}\right\}.$$

We can now use the lower bound on $d_t^{\mathsf{b}}(u, ao)$ in the case $\widehat{p}(uao) \geq \psi$ to achieve the following bound:

$$N \geq \frac{10\overline{U}AO\overline{C}\log(8HUAO/\delta_0)}{\varepsilon}\left(\frac{128\log(AOR)^H}{\mu_0^2} + 31/6\right) + \frac{160\overline{U}AO\overline{C}\log(8HUAO/\delta_0)}{3\varepsilon}$$

$$= \widetilde{\mathcal{O}}\left(\frac{H\overline{U}AO\overline{C}\log(1/\delta)}{\varepsilon\mu_0^2}\right).$$

## C.5 Proof of Theorem 3

Let $\text{TESTDISTINCT}_{\mathcal{X}}$ be the version of $\text{TESTDISTINCT}$ that uses the statistical test in the theorem. The proof of Theorem 3 is achieved by proving two lemmas analogous to Lemmas 11 and 12.

**Lemma 16.** *For $t \in [\![0, H]\!]$, let $\mathcal{Z}_1$ and $\mathcal{Z}_2$ be multisets sampled from distributions $p_1$ and $p_2$ on $\Delta(\Gamma^{H-t})$, and let $\widehat{p}_1$ and $\widehat{p}_2$ be empirical estimates of $p_1$ and $p_2$ due to $\mathcal{Z}_1$ and $\mathcal{Z}_2$, respectively. Under event $\mathcal{E}_{\mathcal{X}}$, if $p_1 = p_2$ then $\text{TESTDISTINCT}_{\mathcal{X}}(t, \mathcal{Z}_1, \mathcal{Z}_2, \delta)$ answers false.*

*Proof.* Due to event $\mathcal{E}_{\mathcal{X}}$ and the triangle inequality we have

$$L_{\mathcal{X}}(\widehat{p}_1, \widehat{p}_2) \leq L_{\mathcal{X}}(\widehat{p}_1, p_1) + L_{\mathcal{X}}(p_1, p_2) + L_{\mathcal{X}}(p_2, \widehat{p}_2)$$
$$\leq \sqrt{\frac{\log(2|\mathcal{X}|/\delta)}{2|\mathcal{Z}_1|}} + 0 + \sqrt{\frac{\log(2|\mathcal{X}|/\delta)}{2|\mathcal{Z}_2|}} \leq \sqrt{\frac{2\log(2|\mathcal{X}|/\delta)}{\min(|\mathcal{Z}_1|, |\mathcal{Z}_2|)}}.$$

This is precisely the condition for which $\text{TESTDISTINCT}_{\mathcal{X}}$ returns false. $\square$

**Lemma 17.** *For $t \in [\![0, H]\!]$, let $\mathcal{Z}_1$ and $\mathcal{Z}_2$ be multisets sampled from distributions $p_1$ and $p_2$ on $\Delta(\Gamma^{H-t})$, and let $\widehat{p}_1$ and $\widehat{p}_2$ be empirical estimates of $p_1$ and $p_2$ due to $\mathcal{Z}_1$ and $\mathcal{Z}_2$, respectively. Under event $\mathcal{E}_{\mathcal{X}}$, if $p_1 \neq p_2$ then $\text{TESTDISTINCT}_{\mathcal{X}}(t, \mathcal{Z}_1, \mathcal{Z}_2, \delta)$ answers true if $\mathcal{Z}_1$ and $\mathcal{Z}_2$ satisfy $\min(|\mathcal{Z}_1|, |\mathcal{Z}_2|) \geq 8\log(2|\mathcal{X}|/\delta)/\mu_0^2$.*

*Proof.* Due to event $\mathcal{E}_{\mathcal{X}}$ and the triangle inequality we have

$$L_{\mathcal{X}}(\widehat{p}_1, \widehat{p}_2) \geq L_{\mathcal{X}}(p_1, p_2) - L_{\mathcal{X}}(\widehat{p}_1, p_1) - L_{\mathcal{X}}(p_2, \widehat{p}_2)$$
$$\geq \mu_0 - \sqrt{\frac{\log(2|\mathcal{X}|/\delta)}{2|\mathcal{Z}_1|}} - \sqrt{\frac{\log(2|\mathcal{X}|/\delta)}{2|\mathcal{Z}_2|}}$$
$$\geq \mu_0 - \sqrt{\frac{2\log(2|\mathcal{X}|/\delta)}{\min(|\mathcal{Z}_1|, |\mathcal{Z}_2|)}} \geq \mu_0 - \sqrt{\frac{\mu_0^2}{4}} = \frac{\mu_0}{2} \geq \sqrt{\frac{2\log(2|\mathcal{X}|/\delta)}{\min(|\mathcal{Z}_1|, |\mathcal{Z}_2|)}},$$

where we have used the condition on $\min(|\mathcal{Z}_1|, |\mathcal{Z}_2|)$ in the lemma. This is precisely the condition for which $\text{TESTDISTINCT}_{\mathcal{X}}$ returns true. $\square$

The remainder of the proof of Theorem 3 is analogous to that of Theorem 2. To satisfy the conditions on $|\mathcal{Z}(uao)|$ in Lemma 17 and Corollary 9, we impose a bound $M_{\mathcal{X}} = 16\log(4/\delta)\log|\mathcal{X}|/\mu_0^2$. We can now choose $b = 16\log|\mathcal{X}|/\mu_0^2$ in Lemma 10 to obtain a bound

$$N \geq \max_{uao} \left\{ \frac{\log(4/\delta)}{d_t^{\text{b}}(u, ao)} \left( \frac{32\log|\mathcal{X}|}{\mu_0^2} + 31/6 \right) \right\}.$$

Since $\text{TESTDISTINCT}_{\mathcal{X}}$ answers correctly under event $\mathcal{E}_{\mathcal{X}}$, Lemma 13 directly applies, with the only difference that we do not need event $\mathcal{E}_{CMS}$ to hold, and hence it is sufficient to choose $\delta_0 = \delta/2UAO$ to ensure that events $\mathcal{E}_{\mathcal{X}}$ and $\mathcal{E}_B$ hold. Using the lower bound on $d_t^{\text{b}}(u, ao)$ yields

$$N \geq \frac{C_{\mathbf{R}}^* \log(8UAO/\delta_0)}{d_{\mathsf{m}}^*} \left( \frac{32\log|\mathcal{X}|}{\mu_0^2} + 31/6 \right) = \widetilde{\mathcal{O}} \left( \frac{C_{\mathbf{R}}^* \log(1/\delta)\log|\mathcal{X}|}{d_{\mathsf{m}}^* \cdot \mu_0^2} \right),$$

We also prove an analogous theorem for the approximation algorithm ADACT-H-A:

**Theorem 18.** *ADACT-H-A$(\mathcal{D}, \delta, \varepsilon, \overline{U}, \overline{C})$ returns an $\frac{\varepsilon}{2}$-approximate RDP $\mathbf{R}'$ with probability at least $1 - 2AOU\delta$ when using the language metric $L_{\mathcal{X}}$ to define a statistical test*

$$L_{\mathcal{X}}(\mathcal{Z}_1, \mathcal{Z}_2) \geq \sqrt{2\log(2|\mathcal{X}|/\delta)/\min(|\mathcal{Z}_1|, |\mathcal{Z}_2|)},$$

*and the size of the dataset $\mathcal{D}$ is at least*

$$|\mathcal{D}| \geq \widetilde{\mathcal{O}} \left( \frac{\overline{U}AO\overline{C}\log(1/\delta)\log|\mathcal{X}|}{\varepsilon\mu_0^2} \right),$$

The proof is also analogous to that of Theorem 14. Concretely, Lemma 15 still holds as long as we ensure $M \geq M_{\mathcal{X}}$ for candidate states such that $\widehat{p}(uao) \geq \psi$, and $N \geq 16 \log(4/\delta)/\psi$ otherwise. Applying Lemma 10 directly yields the following bound on the sample complexity:

$$N \geq \max_{uao} \left\{ \frac{\log(4/\delta)}{d_t^{\mathsf{b}}(u, ao)} \left( \frac{32 \log |\mathcal{X}|}{\mu_0^2} + 31/6 \right) + \frac{16 \log(4/\delta)}{\psi} \right\}.$$

Using the lower bound on $d_t^{\mathsf{b}}(u, ao)$ in the case $\widehat{p}(uao) \geq \psi$ yields

$$N \geq \frac{10\overline{U} AO\overline{C} \log(8UAO/\delta_0)}{\varepsilon} \left( \frac{32 \log |\mathcal{X}|}{\mu_0^2} + 31/6 \right) + \frac{160\overline{U} AO\overline{C} \log(8UAO/\delta_0)}{3\varepsilon}$$

$$= \widetilde{\mathcal{O}} \left( \frac{\overline{U} AO\overline{C} \log(1/\delta) \log |\mathcal{X}|}{\varepsilon \mu_0^2} \right).$$

# D DETAILS OF THE EXPERIMENTS

In this appendix we describe each domain in detail and include examples of RDPs learned by ADACT-H.

## D.1 CORRIDOR

This RDP example was introduced by Ronca & De Giacomo (2021). The environment consists of a $2 \times m$ grid, with only two actions $a_0$ and $a_1$ which moves the agent to states $(0, i+1)$ and $(1, i+1)$ respectively from state $(\cdot, i)$. The goal of the agent is to avoid an enemy which is present in position $(0, i)$ with probability $p_i^0$, and at $(1, i)$ with probability $p_i^1$. The agent receives a reward of $+1$ for avoiding the enemy at a particular column, and the probabilities $p_i^0$ and $p_i^i$ are switched every time it encounters the enemy. When the agent reaches the last column, its position is reset to the first column. The observation space is given by $(i, j, e)$, where $i, j$ is the cell position of the agent and $e \in \{enemy, clear\}$ denotes the presence of the guard in the current cell. Fig 3 shows the minimal automaton obtained by all three algorithms for $H = 5$.

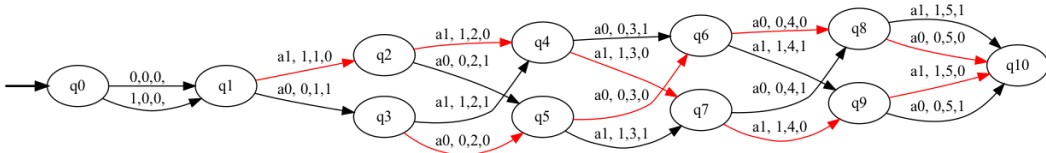

Figure 3: Automaton obtained from the corridor environment. The edges are labelled as [*action, observation, enemy*].

## D.2 T-MAZE(C)

The T-maze(c) environment was introduced by Bakker (2001) to capture long term dependencies with RL-LSTMs. As shown in Figure 1, at the initial position $S$, the agent receives an observation $X$, depending on the position of the goal state $G$ in the last column. The agent can take four actions, *North*, *South*, *East* and *West*. The agent receives a reward of $+4$ on taking the correct action at the T-junction, and $-1$ otherwise, terminating the episode. The agent also receives a $-1$ reward for standing still. At the initial state the agent receives observation 011 or 110, 101 throughout the corridor and 010 at the T-junction. Figure 1 shows the optimal automaton obtained when the available actions in the corridor are restricted to only $East$ (the automaton obtained without this restriction is shown in Figure 6). Table 1 shows our results with the unrestricted action space using the language set $\mathcal{X}_{3,1}$. Both our approaches find the optimal policy in this case, unlike FlexFringe which fails to capture this long term dependency.

Note that T-maze(c) differs from the T-maze of Example 1 in a small but relevant detail: in T-maze(c) the agent deterministically receives the observation 101 when it is in the corridor, but in the T-Maze

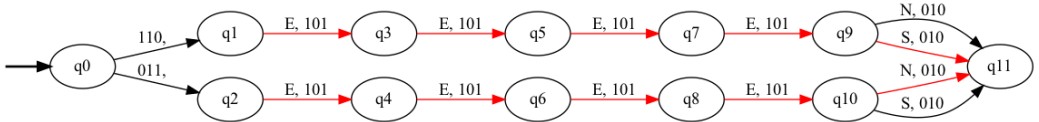

Figure 4: Automaton obtained from T-maze(c) environment with restricted actions. The edges are labelled as [*action*, *observation*].

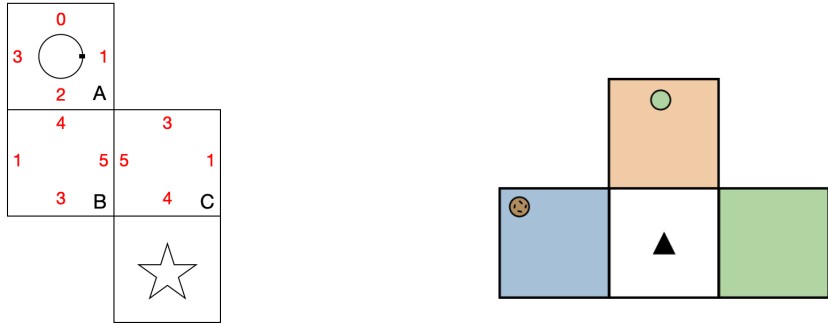

(a) Mini-hall environment (Littman et al., 1995).

(b) Simplified cookie domain.

Figure 5: Environments.

of Example 1 the agent can receive 101 or 111 uniformly at random. The additional observation 111 is a distraction for the agent, rather than a help, since it does not convey any additional useful information.

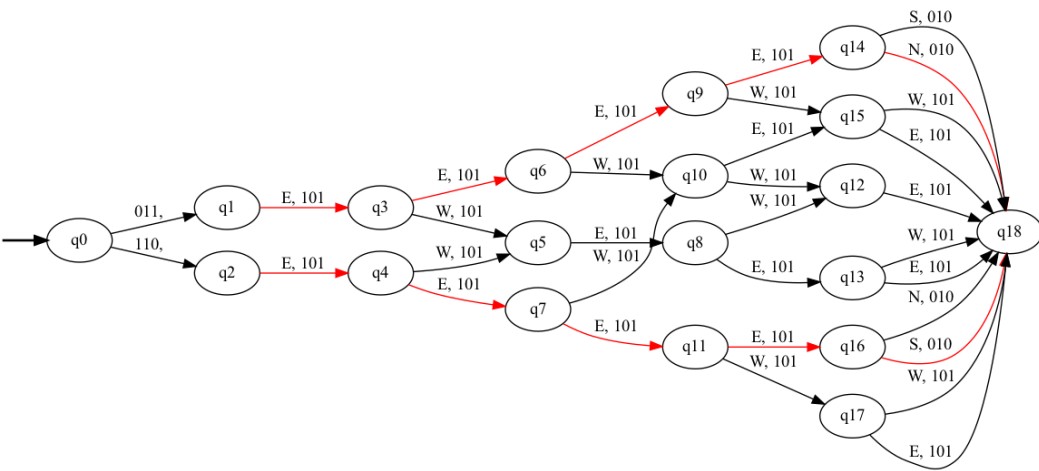

Figure 6: Automaton obtained from T-maze(c) (partially restricted action space), restricted language.

### D.3 COOKIE DOMAIN

We modify the original *cookie domain* as described in (Toro Icarte et al., 2019), to a simpler domain consisting of 4 rooms, *blue*, *white*, *green* and *red* as shown in Fig. 5b. If the agent presses the button in room *red*, a cookie appears in room *blue* or *green* with equal probability. The agent can move *left*, *right*, *up* or *down*, can *press* the button in room *red*, and *eat* the cookie to receive a reward 1, and then it may press the button again. There are 6 possible observations (4 for each room, and 2 for observing the *cookie* in the two rooms). We use the set $\mathcal{X}_{3,1}$ for distinguishability in the restricted language case. Our restricted language approach here finds the optimal policy and the smallest state space.

### D.4  CHEESE MAZE

Cheese maze (McCallum, 1992) consists of 10 states, and 6 observations, and 4 actions. After reaching the goal state, the agent receives a reward of $+1$, and the position of the agent is reinitialised to one of the non-goal states with equal probability. For a horizon of 6, the results for the restricted language (using the language set $\mathcal{X}_{3,1}$) and FlexFringe are comparable, however upon further increasing the horizon, FlexFringe outperforms ADACT-H by learning cyclic RDPs which is not possible in our approach.

### D.5  MINI-HALL

The mini-hall environment (Littman et al., 1995) shown in Fig 5a has 12 states, 4 orientations in 3 rooms, a goal state given by a star associated with a reward of $+1$, 6 observation and 3 actions, and the position of the agent is reset after the goal is reached. This setting is much more complex than the others because 12 states are mapped into 6 observations; for example, starting from observation 3, 3 actions are required under the optimal policy to solve the problem if the starting underlying state was in Room B or C. We use the language set $\mathcal{X}_{3,1}$ and although we get a much larger state space, our algorithm gets closer to the optimal policy. However our CMS approach is not efficient in this case and exceeds the alloted time budget of 1800 seconds, as it requires to iterate over all possible trajectories.

