# OpenReview forum: "Offline RL in Regular Decision Processes: Sample Efficiency via Language Metrics"
_ICLR.cc/2025/Conference — ICLR 2025 Poster_

### Official Review · Reviewer_YHdK · 2024-10-24

**Soundness:** 3
**Presentation:** 4
**Contribution:** 3
**Rating:** 8
**Confidence:** 3

**Summary:**

The paper offers approaches for offline RL in Regular Decision Processes.  It focusses on improving sample efficiency for the state merging part of existing algorithms.  One contribution is a technique that employs a language metric for state comparisons.  The metric is somewhat akin to the principles behind e.g. graph kernels based on walks (see Gaertner, Flach, Wrobel, 2003) as it compares the probability distributions of the set of traces that can be generated by starting in two given states (nodes) and walking around in the RDP (on the graph).  The paper presents an efficient way to find relevant (i.e. state distinguishing) patterns in historic traces and a memory saving approach for storing probability distributions over large histories.  The paper includes a theoretical analysis and a (brief) experimental study.

**Strengths:**

The contributions are well placed in the context of directly related work.  Although some more distant related work might have been missed, such as the graph kernel work mentioned in the summary, this represents only a minor point.  To the best of my knowledge, the paper contains sufficient novelty to represent a good contribution, and it is nice to see the domain of tractable RL algorithms extended to settings beyond MDPs.
The work and analysis presented are substantial.  I will honestly admit I did not rigorously go through all the mathematics of the paper and the appendices.  However, the parts I did go through illustrated the care for detail required.
The paper is very well written too.  There is a lot of material to get through and the organisation of the paper is well done and the focus is good.  This may also explain the limited experimental support supplied in the paper.  The authors clearly decided to focus on the theoretical guarantees they could provide and supplied a (in their own words) numerical experimental evaluation.

**Weaknesses:**

Being unfamiliar with the term Regular Decision Process, I had to dig up the 2024 paper (there is also a 2019 one, which I missed) in the hope of finding more about the type of domains they represent.  The authors do define the RDP setting, but the paper does not succeed in relating the usefulness of the setting, i.e. it remains unclear how widely applicable the presented techniques are.  It would have been nice to see some less theoretical application domains that fit the setting, are non-trivial to translate/transform into MDPs, yet present an interesting RL challenge.  It is very hard for the reader to come up with one of these themselves, or at least it is in my experience while reading the paper.

**Questions:**

I understand it is always easy to ask for more, but your paper would substantially benefit from a non-toy environment, both mentioned as a fit for the learning setting you are tackling, but of course also used as an illustrative experiment to show what your contributions make possible that wasn't before.  RL research's biggest boost came from showing solutions of non-trivial environments.  I'm afraid that your good work might be lost without such a demonstrator.

---

> ### Author Response · Authors · 2024-11-21
>
> We thank the reviewer for the detailed review and valuable feedback, which we will integrate into the paper. We discuss several points meant to address weaknesses and questions in the review.
>
> **Example application domains** : As mentioned in the answer to other reviewers, the complexity of the RDP directly impacts the sample efficiency of RDP learning. In turn, since the RDP states are sufficient to predict the future, the difficulty of learning the RDP is directly linked to the difficulty of translating a generic episodic decision process into an MDP. We believe that the language metric should make it possible to learn simple RDPs independently of the number of observations and actions. This is possible precisely when the amount of information to remember about the past is relatively limited.
>
> **Non-trivial environment** : We strongly believe that the successful application of reinforcement learning to non-trivial environments was preceded by many important theoretical results. An important objective of the present paper is to illustrate the benefits of the proposed language metric, and we believe that the experiments supplement the theory by showing its superior performance in several domains. Of course this is only a first step towards handling larger domains.

---

> > ### Comment · Reviewer_YHdK · 2024-11-28
> >
> > Thank you for your reply.  I am not sure I made myself fully clear about the weakness I raised, since I don't understand how your reply addresses it.  What I think misses from an otherwise good paper that I recommend acceptance for, is a clear description of how the RDP setting extends the MDP setting with respect to everyday(?) RL suitable tasks.  Could you provide an idea about the applicability of the presented techniques by providing 1 or 2 examples of real world domains or applications where RDPs are particularly useful or necessary over the standard MDP setting?

---

> > > ### Author Response · Authors · 2024-11-29
> > >
> > > We thank the reviewer for clarifying the question. Below we provide an additional response.
> > >
> > > At a high level, RDPs are useful in all domains in which in order to act optimally the agent must rely on its history of interaction with the environment. This is in contrast with MDPs, which assume that the future is conditionally independent from the past given the current state.
> > > Many examples can be found in the literature. Gabaldon [1] presents an environment where a robot is working in a biological research facility with various zones having different safety-levels.The robot is considered to be contaminated if it has touched an hazardous material, and the effect of the action *touch* on a new material will depend on its history of interaction in the environment (whether or not the robot has visited a disinfection station in-between). Or the robot cannot *open* the entrance of a particular lab, if the temperature of that lab exceeded a particular temperature since its last entry in the lab.
> > >
> > > Ni et al.[2] present many of these environments with temporal dependencies, for example,
> > > * Goal Navigation [3] environment where the agent needs to memorize crucial information throughout the episode in order to optimally reach the goal state.
> > > * PsychLab [3] environments simulating psychology laboratories like Arbitrary Visuomotor Mapping where objects are presented in a series to an agent, and each object is associated with a look-direction. If the agent looks in the associated direction the next time it sees a particular object in the episode, it receives a positive reward.
> > > * Spot the Difference [3] environment, where the agent has to move between two rooms, with a “delay” corridor in between and correctly identify the difference in the two rooms.
> > > * Memory Maze [4], where the agent is placed in a maze with $K$ colored balls, and the agent must accumulate the balls in a specific order (which is randomly set for each episode). To act optimally here, the agent needs to memorize the position of objects, wall layouts etc.
> > >
> > > Although in real life similar tasks will have a much larger observation space, whenever the underlying transition and reward functions are *regular*, they can be modelled as an RDP that can be learned using the methods we propose. Such problems could be alternatively modelled as POMDPs, but the class of RDPs is a strict subset of the class of POMDPs [5], so RDP learning methods have the potential to be significantly more efficient than POMDP learning methods.
> > >
> > > In our paper we focus on the T-maze as a running example because it exemplifies the essence of environments where the history must be taken into account. There, in order to act optimally, the agent must remember the goal location which is disclosed at the beginning of an episode. The environment is purposefully simple, but such a simple dependence on the history can appear in real-world domains that are arbitrarily complex.
> > >
> > > For instance, it can appear in robotics applications. Say a robot must navigate an arbitrarily large grid, it starts at the bottom left corner, and it is immediately told whether the goal is in the top right corner or in the bottom right corner. The grid itself can be arbitrarily complex, e.g., with obstacles to avoid. Still, in order to act optimally, the robot must rely on the information present in the history—i.e., the initial piece of information disclosing the location of the goal.
> > > Also in the domains we consider in the experiments it is key to rely on the history of past events, and they are relevant for applications in robotics.
> > >
> > > [1] Gabaldon, A. (2011). Non-markovian control in the situation calculus. Artificial Intelligence, 175(1), 25-48.
> > >
> > > [2] Ni, Tianwei & Ma, Michel & Eysenbach, Benjamin & Bacon, Pierre-Luc. (2023). When Do Transformers Shine in RL? Decoupling Memory from Credit Assignment. 10.48550/arXiv.2307.03864.
> > >
> > > [3] Fortunato, M., Tan, M., Faulkner, R., Hansen, S.S., Badia, A.P., Buttimore, G., Deck, C., Leibo, J.Z., & Blundell, C. (2019). Generalization of Reinforcement Learners with Working and Episodic Memory. ArXiv, abs/1910.13406.
> > >
> > > [4] Pašukonis, J., Lillicrap, T.P., & Hafner, D. (2022). Evaluating Long-Term Memory in 3D Mazes. ArXiv, abs/2210.13383.
> > >
> > > [5] Brafman, r. & De Giacomo, G. (2024). Regular decision processes. Artificial Intelligence 331, 104113.

---

### Official Review · Reviewer_9Ta3 · 2024-11-02

**Soundness:** 4
**Presentation:** 3
**Contribution:** 3
**Rating:** 8
**Confidence:** 4

**Summary:**

The authors present a new metric $L_\chi$ regular decision processes (RDP) based on formal languages. In theorem 1, this metric is shown to be an improvement on previous metrics $L_q^p$ by outlining a family of RDPs for which the  $L_q^p$-distinguishability decays as time horizon increases while the $L_\chi$-distinguishability is constant. Example 4 is particularly useful in illustrating this phenomenon where a T shaped grid world with an increasing corridor length is causing a decay of distinguishability while it is constant in the second case.  They then compare the distinguishability of these metrics for the policies produced by the algorithm A DACT-H.

**Strengths:**

Well written. The example of the T shaped world help fix the cause of the deficiency of the $L_q^p$ metric and how the $L_\chi$ fixes the problem.

**Weaknesses:**

The proof of theorem 1 looks correct but I did not have time to check all details. A quick outline of how it works would be welcome.

**Questions:**

I would like to see a definition of "episode trace". I think I understand the meaning from the context but it would be best to clearly write it down somewhere.
The ADACT algorithm is first mentioned on line 434 but no definition is given before that. I later found the definition in the appendix but it would be good to have a link to that definition when it is first mentioned.

---

> ### Author Response · Authors · 2024-11-21
>
> We thank the reviewer for the detailed review and valuable feedback, which we will integrate into the paper. We discuss several points meant to address weaknesses and questions in the review. In particular, the question is addressed by points “Episode traces” and “AdaCT-H”.
>
> **Episode traces** : We use the terms "episode", "trace" and "suffix" interchangeably in the paper, and all refer to sequences of action-reward-observation triplets introduced at the beginning of Section 2.3. We have clarified the use of terminology in the paper.
>
> **Outline of Theorem 1** : In the T-maze introduced in Example 3, while traversing the corridor each observation is random ($101$ or $111$). Hence in a corridor of length $H-1$, there are $2^{H-1}$ possible observation sequences, all equally likely. At the end of the corridor, the random reward depends on the first observation ($110$ or $011$). Hence the $L_\infty$ norm has to compare the difference in probability of traces of length $H$, each of which has probability $1/2^H$. The distinguishability is equal to the maximum difference in probability, which can be at most $1/2^H$ in case the distribution on final rewards is different. On the other hand, the language metric only compares whether or not a certain final reward occurs, and each reward has probability $1/2$, so the distinguishability is $1/2$ in this case.
>
> **AdaCT-H** : The algorithm is first introduced in Section 3, but we have added a reference to Appendix A when AdaCT-H is again mentioned in Section 4.2.

---

### Official Review · Reviewer_92xy · 2024-11-04

**Soundness:** 3
**Presentation:** 2
**Contribution:** 2
**Rating:** 6
**Confidence:** 3

**Summary:**

This paper explores offline reinforcement learning (RL) within Regular Decision Processes (RDPs), a subclass of non-Markovian environments. The authors propose improvements on existing methods through a novel language metric and the use of Count-Min Sketch (CMS) to enhance sample efficiency and reduce memory costs. They provide theoretical guarantees and some empirical results in specific domains, comparing their approach to the state-of-the-art methods.

**Strengths:**

The paper offers a theoretical framework for a new language metric and CMS integration in offline RL for RDPs. The analysis of PAC sample complexity for RDPs provides valuable insight into distinguishing state representations in non-Markovian settings.

**Weaknesses:**

1. The primary contribution, a novel language metric, builds heavily on established language hierarchy concepts and does not provide a groundbreaking shift in methodology. CMS is also a well-established technique, and its application here does not significantly differentiate the approach from prior state-merging algorithms.  I think the author should highlight the difference and the technical contribution.
2. Experimental results are limited to a small set of benchmark domains that are relatively simple, and the benchmark algorithms used for comparison are limited. More famous RDP algorithms might be involved like Omega-RDP (Hahn et al., 2023) and Grid-world (Lenaers and Otterlo, 2021).

---
Hahn, E. M., Perez, M., Schewe, S., Somenzi, F., Trivedi, A., & Wojtczak, D. (2024, March). Omega-Regular Decision Processes. In Proceedings of the AAAI Conference on Artificial Intelligence (Vol. 38, No. 19, pp. 21125-21133).

Lenaers, N., & van Otterlo, M. (2022). Regular decision processes for grid worlds. In Artificial Intelligence and Machine Learning: 33rd Benelux Conference on Artificial Intelligence, BNAIC/Benelearn 2021, Esch-sur-Alzette, Luxembourg, November 10–12, 2021, Revised Selected Papers 33 (pp. 218-238). Springer International Publishing.

**Questions:**

1. Given the reliance on existing language hierarchy theories, how does the proposed language metric meaningfully differ in terms of its theoretical impact on distinguishing states within RDPs?
2. Can the authors expand on any potential limitations of CMS in larger, more complex RDPs, especially concerning long planning horizons?

---

> ### Author Response · Authors · 2024-11-21
>
> We thank the reviewer for the detailed review and valuable feedback, which we will integrate into the paper.
> We discuss several points meant to address weaknesses and questions in the review. In particular, the first question is addressed mainly by the point “Novelty of the language metric”. The second question is addressed mainly by the point “Count-Min-Sketch”.
>
> **Novelty of the language metric** : Contrary to the weakness perceived by the reviewer, we claim that the proposed language hierarchy does not rely on existing language hierarchies. The only connection with existing classes of languages is the operator in Definition 1, which is only loosely inspired by classes of languages in the first level of the dot-depth hierarchy. Thus, our contribution with regards to the language metric and the language hierarchies is entirely novel. Furthermore, we believe that using a language-theoretic approach to prove sample complexity guarantees in a reinforcement learning setting is extremely innovative.
>
> **Count-Min-Sketch** : Though CMS is indeed used in other automaton learning algorithms (e.g. Baumgartner & Verwer[1]), other authors do not provide an extensive statistical analysis, and hence the choice of CMS parameters appears arbitrary. In contrast, our analysis sheds light on how to properly choose the CMS parameters, and as far as we know, we are the first to prove sample complexity guarantees for CMS in the automaton learning setting. As discussed in the answer to Reviewer K6iR, the difficulty of learning complex RDPs is not due to CMS, but rather due to properties of the RDP itself.
>
> **Omega-regular decision processes (ODPs)** : As far as we can tell, ODPs are a generalization of RDPs, and hence ODPs should be at least as hard to learn as RDPs. The paper on ODPs present several *classical* complexity results (e.g. EXPTIME-hardness), while one of our major contributions is to provide a *statistical* analysis of the RDP learning algorithm and its sample complexity.
>
> **Limitation of benchmarks** : As a primarily theoretical RL paper, the key goal is to devise provably sample-efficient algorithms that admit meaningful sample complexity bounds. The motivation behind providing numerical experiments here is: (1) to provide further insights into the presented results, thereby enriching discussion; and (2) to showcase that the presented approaches are not merely high-level theories and can indeed be implemented. We believe the domains used in our experiments, albeit rather small, render quite useful in  achieving (1) and (2). Simple domains appear especially effective to spell out some key quantities (e.g., distinguishability) that arise in the context of RDP.  We believe conducting experiments on larger domains fall into the scope of a follow-up work, where one may slightly depart from the theoretical framework via applying some techniques that are yet justifiable from a practical standpoint or may use some domain knowledge. Finally, we stress that releasing the code – which we plan to do – is a plus and paves the way to examining the presented approaches on larger domains.
>
> **References** : We plan to incorporate the references mentioned by the reviewer in the final version of the paper.
>
> [1] Baumgartner, Robert, and Sicco Verwer. "Learning state machines from data streams: A generic strategy and an improved heuristic." International Conference on Grammatical Inference. PMLR, 2023.

---

> > ### Comment · Reviewer_92xy · 2024-12-02
> >
> > I thank the authors for the detailed reply, which solves most of my concerns. I raise the score to 6.

---

### Official Review · Reviewer_K6iR · 2024-11-04

**Soundness:** 3
**Presentation:** 2
**Contribution:** 3
**Rating:** 6
**Confidence:** 4

**Summary:**

This paper considers the offline reinforcement learning (RL) problem when the offline trajectories data are non-Markov---precisely when they are generated by an underlying Regular Decision Process (RDP). To address this problem, they propose two novel algorithmic adaptations of ADACT-H (Cipollone et al., 2023) from prior work to learn the underlying RDP: one using the Count-Min Sketch (CMS) and another using a novel language metric ($L_X$) for efficient learning of episodic RDPs from offline data. The language metric is shown to be a generalization of previous ones like $L_1$ and $L_{\infty}$, capturing desired properties of both to improve state distinguishability. They show that the proposed approaches significantly reduce sample and space complexity compared to prior methods. Finally, they conduct experiments across five domains from prior works to demonstrate the proposed methods' advantages in terms of runtime, number of states, and reward maximization.

**Strengths:**

- The proposed language metric is novel and offers a unique approach to state distinguishability, unifying and extending traditional distance metrics (like $L_1$ and $L_{\infty}$). This potentially sets a new standard for offline RDP learning.
- This is primarily a theoretical paper. The proposed approach has clearly stated assumptions and is supported by rigorous sample complexity analysis with proofs. Interestingly, through their theoretical analysis, the authors also uncover a mistake in one of theoretical results from the ADACT-H (Cipollone et al., 2023) paper. Such results are very much appreciated. Although the experiments are in fairly simple small environments with a single baseline, their inclusion is also appreciated as it helps better understand the applicability of the proposed approach.
- The paper is generally well-written, with somewhat clear explanations of the RDP framework and experimental setup. Although some sections (e.g., language metric details) could benefit from further simplification for broader accessibility. They also provide pseudocodes in the appendix for better clarity.
- This work contributes meaningfully to offline RL in non-Markov environments, where observations and rewards depend on past transitions.

**Weaknesses:**

- The method seems to only be applicable to very small environments in practice, given that it attempts to learn the underlying RDP for both obsservation and reward transitions. This doesn't look like it can be applied to larger environments (such as high-dimensional observations and continuous crontrol ones), unlike prior works in the online setting like Toro Icarte et al., 2019.

- The approach assumes $\pi^b$ maintains a non-zero minimum distinguishability, which may limit the algorithm's application in environments with varying policy behaviors or unobservable state spaces. Further discussion on generalizing this assumption would enhance the paper's robustness.

- The CMS-based approach suffers exponential complexity with the horizon \(H\), which could impact performance in long-horizon tasks, as evidenced in the Mini-hall domain experiment. Addressing this limitation or providing recommendations on CMS applicability range would be beneficial.

- The sample complexity bound relies on a parameter inversely related to the minimum occupancy of the optimal policy $d_m^*$, which could become infeasible in scenarios with low-reachability states. Including strategies or adjustments for low-occupancy environments could further strengthen the results.

- In general, the paper is very hard to follow for an RL researcher (at least it was for me). This is partially because of too many notations, and a mismatch of notations from different fields. For example, some come from the RL literature (like the value function $V(.)$) and others like $Q(.)$ which usually represents the action-value function in RL now represents the set of RDP states (instead of $S$ or even $U$). I think a lot of work needs to be done to simplify the notations in the background and improve the clarity of Section 4.

**Questions:**

- Could the authors discuss possible modifications to their approach if $\mu_0$ is close to zero in some states? What would the practical implications be in such cases?
- Can the proposed approach be applied to settings where the RDP structure is much larger, high dimensional, and even continuous? E.g. [1].

[1] Allen, Cameron, et al. "Mitigating Partial Observability in Sequential Decision Processes via the Lambda Discrepancy." arXiv preprint arXiv:2407.07333 (2024).

---

> ### Author Response · Authors · 2024-11-21
>
> We thank the reviewer for the detailed review and valuable feedback, which we will integrate into the paper. We discuss several points meant to address weaknesses and questions in the review. In particular, the first question is addressed mainly by the point “Assumption on distinguishability” and “Complexity of RDPs”. The second question is addressed mainly by the point “Continuous observations and actions”.
>
> **Comparison to the reward machine literature** : We remark that unlike Toro Icarte et al., we do not assume access to an existing label set and label function, and instead learn RDPs directly from a dataset of action-observation-reward episodes. We believe this problem to be significantly harder, and at the same time more realistic in case a label function is hard to define manually. Hence the two approaches are not directly comparable, and the RDPs we obtain are generally larger since we cannot exploit the compression afforded by the labels.
>
> **Scalability** : An interesting direction for future research would be to assume that the underlying MDP state is the cross-product of the observation and the automaton state, i.e. $S = \mathcal{O} \times \mathcal{U}$ in our new notation. This is generally done in the reward machine literature, and makes the automaton part to be learned considerably more compact. A significant challenge is to achieve such an extension while still maintaining formal sample complexity guarantees.
>
> **Assumption on distinguishability** : Cipollone et al. prove a lower bound on the sample complexity which is inversely proportional to the distinguishability of the $L_1$-norm. Hence the difficulty of the RDP learning problem increases as the distinguishability approaches 0, and we do not believe that we can efficiently learn the exact RDP in this case, or easily generalize Assumption 1 (but perhaps efficient approximation algorithms are possible).
>
> **Minimum occupancy** : The approximation algorithm AdaCT-H-A of Cipollone et al. (which appears in Appendix A) partially alleviates the dependence on the minimum occupancy $d_m^*$. In fact, Cipollone et al. prove a sample complexity bound which excludes $d_m^*$ but in our corrected proof it appears as an additive term. In our view, the most promising direction for removing the dependence on $d_m^*$ (and on the concentrability $C^*$) is to develop sample-efficient *online* algorithms, which is something that we plan to investigate in future work.
>
> **Complexity of RDPs** : Several problem-specific parameters that appear in the sample complexity bound, such as $C^*$, $d_m^*$ and $\mu_0$, strongly depend on the complexity of the RDP. Concretely, if the RDP is reasonably small and has no low-probability branches, then we can control the magnitude of $C^*$ and $d_m^*$. As we show in the paper, we can also often control the magnitude of $\mu_0$ using our novel language metric. Hence the novel language metric should allow us to efficiently learn such well-behaved RDPs, even if the action and observation spaces are large. Since the lower bound of Cipollone et al. includes both $C^*$ and $\mu_0$, learning exact RDPs becomes significantly harder as the complexity of the RDP increases, no matter which algorithm is used.
>
> **Continuous observations or actions** : Since we do not assume access to an existing label set, the only possibility we foresee is to learn a discrete latent representation ("labels") from experience, perhaps using unsupervised learning or another optimization objective, and use such latent variables as transition labels in the learned RDP.
>
> **Count-Min-Sketch** : Note that the exponential complexity in the horizon H is due to the distinguishability of the $L_\infty$-norm, which is problem-dependent. Hence the exponential complexity is not due to CMS (which always improves the memory complexity), but instead due to the problem being solved and the metric used. This is precisely the limitation addressed by the language metric, which can unfortunately not be combined with CMS while maintaining strong sample complexity guarantees.
>
> **Notation**  : We have changed our notation for the set of RDP states from $\mathcal{Q}$ to $\mathcal{U}$ according to the suggestions of the reviewer.

---

> > ### Comment · Reviewer_K6iR · 2024-12-02
> >
> > Thank you to the authors for their detailed response. It has helped me understand the work better and increased my confidence in my score.
> >
> > As a side remark, what the authors mentioned for 'Continuous observations or actions" can also be applied to learn a labeling function for reward machines. Such a discretisation of the observation space makes more sense for learning/representing the reward function as a reward machine (since the reward function is generally sparse without reward engineering). It makes less sense for learning/representing the whole MDP as an RDP, since that will have important implications on the size of the RDP and whether or not the transition dynamics are Markov.

---

### Meta-Review · Area_Chair_qtfR · 2024-12-23

**Metareview:**

This work studies offline RL in a special class of non-Markovian environments --- RDPs, where hidden finite-state automata capture dependencies between past interactions and future outcomes. It introduces two novel techniques: a formal language theory-based metric, which improves sample efficiency, and a Count-Min-Sketch approach that reduces memory requirements for long planning horizons. The authors also provide PAC sample complexity bounds and experimental validation to support their approach. The reviewers unanimously agree that this paper makes an interesting and valuable contribution to the field. As such, they all recommend accepting the paper.

**Additional Comments On Reviewer Discussion:**

The reviewers raised questions regarding the novelty of the work and certain technical details, which were thoroughly addressed in the rebuttal. With all reviewers providing positive scores, I recommend acceptance.

---

### Decision · Program_Chairs · 2025-01-22

Accept (Poster)